# SEC24A deficiency lowers plasma cholesterol through reduced PCSK9 secretion

Xiao-Wei Chen[1], He Wang[1], Kanika Bajaj[2], Pengcheng Zhang[2], Zhuo-Xian Meng[1], Danjun Ma[3], Yongsheng Bai[4], Hui-Hui Liu[5], Elizabeth Adams[6], Andrea Baines[6], Genggeng Yu[1], Maureen A Sartor[4], Bin Zhang[5], Zhengping Yi[7], Jiandie Lin[1], Stephen G Young[8], Randy Schekman[9], David Ginsburg[10]*

[1]Life Sciences Institute, University of Michigan, Ann Arbor, United States; [2]Department of Molecular and Cell Biology, University of California, Berkeley, Berkeley, United States; [3]Department of Pharmaceutical Sciences, College of Pharmacy and Health Sciences, Wayne State University, Detroit, United States; [4]Department of Bioinformatics, University of Michigan, Ann Arbor, United States; [5]Department of Molecular Medicine, Cleveland Clinic, Cleveland, United States; [6]Program in Cell and Molecular Biology, University of Michigan, Ann Arbor, United States; [7]Department of Pharmaceutical Sciences, Wayne State University, Detroit, United States; [8]Department of Medicine and Human Genetics, University of California, Los Angeles, Los Angeles, United States; [9]Department of Molecular and Cell Biology, Howard Hughes Medical Institute, University of California, Berkeley, Berkeley, United States; [10]Division of Molecular Medicine & Genetics, Howard Hughes Medical Institute, University of Michigan, Ann Arbor, United States

**Abstract** The secretory pathway of eukaryotic cells packages cargo proteins into COPII-coated vesicles for transport from the endoplasmic reticulum (ER) to the Golgi. We now report that complete genetic deficiency for the COPII component SEC24A is compatible with normal survival and development in the mouse, despite the fundamental role of SEC24 in COPII vesicle formation and cargo recruitment. However, these animals exhibit markedly reduced plasma cholesterol, with mutations in *Apoe* and *Ldlr* epistatic to *Sec24a*, suggesting a receptor-mediated lipoprotein clearance mechanism. Consistent with these data, hepatic LDLR levels are up-regulated in SEC24A-deficient cells as a consequence of specific dependence of PCSK9, a negative regulator of LDLR, on SEC24A for efficient exit from the ER. Our findings also identify partial overlap in cargo selectivity between SEC24A and SEC24B, suggesting a previously unappreciated heterogeneity in the recruitment of secretory proteins to the COPII vesicles that extends to soluble as well as trans-membrane cargoes.

*For correspondence: ginsburg@umich.edu

## Introduction

One-third of the vertebrate genome is predicted to encode proteins that are sorted into the secretory pathway en route to intracellular organelles, the cell surface, or the extracellular space (*Palade, 1975*; *Bonifacino and Glick, 2004*). Following synthesis in the endoplasmic reticulum, trans-membrane and soluble proteins co-translationally inserted into the ER are packaged into transport vesicles coated with COPII (coat protein complex II) for export from the ER and delivery to the Golgi for further processing (*Lee et al., 2004*). The assembly of the COPII coat is initiated at ER exit sites upon activation of the GTPase SAR1, which recruits the inner-coat heterodimer SEC23/SEC24, followed by the

**eLife digest** The endoplasmic reticulum (ER) is a structure that performs a variety of functions within eukaryotic cells. It can be divided into two regions: the surface of the rough ER is coated with ribosomes that manufacture various proteins, while the smooth ER is involved in activities such as lipid synthesis and carbohydrate metabolism. Proteins synthesized by the ribosomes attached to the rough ER are generally transferred to another structure within the cell, the Golgi apparatus, where they undergo further processing and packaging before being secreted or transported to another location within the cell.

Proteins are shuttled from the ER to the Golgi apparatus by vesicles covered with coat protein complex II (COPII). This complex is composed of an inner and outer coat, each of which is assembled primarily with two different SEC proteins: the SEC23/SEC24 protein heterodimer forms the inner coat of the COPII vesicle, and plays a key role in recruiting the appropriate protein cargos to the transport vesicle, while the SEC13/SEC31 protein heterotetramer forms the outer coat and is generally responsible for regulating vesicle size and rigidity.

Previous work found that mammals, including humans and mice, harbor multiple copies of several SEC protein genes, including two copies of *SEC23* and four copies of *SEC24*. Both copies of *SEC23* are derived from the same ancestral gene, and all four copies of *SEC24* are derived from a different ancestral gene, and the availability of these copies potentially expands the range of properties that the vesicles can have. Insight into the roles of each SEC protein has come from work with *SEC* mutants. For example, a mutation in *SEC23A* was found to cause skeletal abnormalities in humans.

Here, Chen et al. report the results of experiments which showed that mice with an inactive *Sec24a* gene could develop normally. However, these mice experienced a 45% reduction in their plasma cholesterol levels because they were not able to recruit and transport a secretory protein called PCSK9, which is a critical regulator of blood cholesterol levels.

The work of Chen et al. reveals a previously unappreciated complexity in the recruitment of secretory proteins to the COPII vesicle and suggests that the various combinations of SEC proteins influence the proteins selected for transport to the Golgi apparatus. The work also identifies *Sec24a* as a potential therapeutic target for the reduction of plasma cholesterol, a finding that could be of interest to researchers working on heart disease and other conditions exacerbated by high cholesterol.

assembly of the outer-coat heterotetramer SEC13/SEC31 to generate carrier vesicles destined for the Golgi (*Lee et al., 2004*; *Gurkan et al., 2006*).

A key sorting event in ER-Golgi transport relies on recognition of specific signals within cargo molecules by the SEC24 subunit of the COPII complex (*Miller et al., 2002*, *2003*; *Lee et al., 2004*), though SEC23 may also contribute to cargo selection in some cases (*Kim et al., 2012*). The SEC31/SEC13 outer-coat regulates the size and rigidity of COPII coats to package specialized cargos (*Copic et al., 2012*; *Jin et al., 2012*). Mammals express multiple paralogous forms of COPII, including two SAR1 GTPases (SAR1A/B), two SEC23s (SEC23A/B), four SEC24s (SEC24A-D), as well as two SEC31s (SEC31A/B), thus expanding the repertoire of potential COPII coat structures (*Wendeler et al., 2007*; *Zanetti et al., 2011*). Biochemical and structural studies of the COPII complex have identified multiple cargo recognition sites on SEC24 (*Bi et al., 2002*; *Miller et al., 2003*; *Bickford et al., 2004*; *Gurkan et al., 2006*; *Mancias and Goldberg, 2008*). The four mammalian SEC24 paralogs can be divided into two subfamilies, SEC24A/B and SEC24C/D, sharing ~60% sequence identity within but only ~25% identity across subfamilies (*Mancias and Goldberg, 2008*; *Zanetti et al., 2011*). Mammalian SEC24A/B exhibit ~30% sequence identity to yeast SEC24p, compared to ~25% for SEC24C/D. The latter share ~30% identity with LST1p or ISS1p, the non-essential SEC24 paralogs in yeast (*Roberg et al., 1999*; *Peng et al., 2000*; *Shimoni et al., 2000*).

Although deletions of SAR1, SEC23, or SEC24 are all lethal in yeast (*Lee et al., 2004*), consistent with the broad function of ER-Golgi transport, mutations in the genes encoding several mammalian COPII components have been associated with remarkably limited phenotypes, often restricted to a specific cell type or tissue. Mutations in human *SAR1B* result in chylomicron retention disease (Anderson Disease), a distinct defect in fat absorption due to reduced chylomicron assembly and secretion by intestinal enterocytes (*Jones et al., 2003*; *Annesi et al., 2007*). In contrast, missense mutations in

human *SEC23A* result in a characteristic syndrome of malformations restricted to the craniofacial skeleton (*Boyadjiev et al., 2006*, *2010*), with similar skeletal abnormalities observed in SEC23A-deficient zebrafish (*Lang et al., 2006*). Mutations in human *SEC23B* result in the autosomal recessive disorder congenital dyserythropoietic anemia type II, with abnormalities restricted to the hematopoietic erythroid compartment (*Bianchi et al., 2009*; *Schwarz et al., 2009*). Surprisingly, although red blood cells appear grossly normal in SEC23B-deficient mice, these animals die at birth due to dramatic destruction of the pancreas (*Tao et al., 2012*).

Though human deficiency has not yet been reported for any of the four SEC24 paralogs, SEC24B deficiency in mice leads to failure in neural tube closure resulting from missorting of the signaling molecule VANGL2 (*Merte et al., 2010*). In contrast, murine SEC24D deficiency results in very early embryonic lethality (*Baines et al., In press*). Here, we report that complete deficiency of SEC24A is compatible with normal development and survival in mice. However, these animals exhibit markedly reduced plasma cholesterol due to selective blockade in the secretion of PCSK9, a circulating factor that negatively regulates cell surface LDL receptor expression.

## Results

### SEC24A null mice are viable and exhibit normal survival and development

SEC24A-deficient mice were generated from an ES cell line carrying a gene trap insertion into intron two of *Sec24a* (*Figure 1A*). The gene trap allele (*Sec24a^gt^*) is predicted to direct expression of a hybrid mRNA fusing *Sec24a* exons 1 and 2 with the gene trap cassette, resulting in a chimeric protein missing the C-terminal ~85% of SEC24A (encoded by exons 3–23). An intercross between *Sec24a^+/gt^* heterozygous mice produced offspring of all three genotypes at the expected Mendelian ratios (*Table 1*). RT-PCR analysis of liver RNA prepared from *Sec24a^gt/gt^* mice showed a >1000 fold reduction in normal splicing from *Sec24a* exon 2 to exon 3 across the gene trap, compared to control wild type mice (*Figure 1B*). Immunoblotting detected an ~50% reduction in SEC24A protein in whole brain protein extracts from heterozygous *Sec24a^+/gt^* mice compared to wild type littermates, with no SEC24A detected in extracts from *Sec24a^gt/gt^* mice (*Figure 1C*). Of note, SEC24B protein levels were increased in brain lysates from *Sec24a^gt/gt^* mice, with a potential slight increase in *Sec24a^+/gt^* heterozygous mice, though no differences were observed for SEC24C or SEC24D (*Figure 1C*).

*Sec24a^gt/gt^* mice were grossly indistinguishable from their wild type littermates and developed normally to adulthood with no difference in body weight on a normal or high fat diet (*Figure 1D*). Kaplan-Meier analysis showed no difference in the survival of wild type and *Sec24a^gt/gt^* mice up to 12 months of age (*Figure 1E*). *Sec24a^gt/gt^* male and female mice both exhibited normal fertility at ~8 weeks of age with normal litter size (7.7 ± 1.4, n = 6, compared to 7.1 ± 1.0, n = 10 for congenic C57BL6/J controls, p>0.35). Gross and routine microscopic survey of multiple tissues failed to identify any obvious morphologic abnormalities in adult *Sec24a^gt/gt^* mice (*Figure 1F*). Complete blood count (CBC) surveys were essentially normal in *Sec24a^gt/gt^* mice compared to wild type controls, except for an ~4% increase in mean corpuscular volume (MCV) and mean corpuscular hemoglobin (MCH) and a compensatory ~6% decrease in red blood cell (RBC) number (*Table 2*).

To exclude a contribution from residual function of the *Sec24a^gt^* allele or a potential 'passenger' gene mutation as a result of gene targeting (*Westrick et al., 2010*), we generated an additional series of *Sec24a* deficient mice from a second, independent *Sec24a* targeted allele (*Figure 2A*). The parental allele (*Sec24a^cgt^*) contains a conditional gene trap insertion in *Sec24a* intron 4. Excision of the LoxP-flanked selection cassette and *Sec24a* exon 5 by *Cre* recombinase generated the *Sec24a^gt2^* allele, which results in a frame shift after the deleted exon 5 in addition to the gene trap. Removal of the FRT-flanked gene trap and selection cassette by *Flpe* recombinase generated the *Sec24a^fl^* allele, which can be converted to the null *Sec24a^-^* allele by *Cre* recombinase. *Sec24a^+/cgt^* or *Sec24a^+/gt2^* intercrosses both produced offspring with all three genotypes at the expected Mendelian ratios (*Table 1*). No SEC24A protein was detected in brain protein extracts from either *Sec24a^cgt/cgt^* or *Sec24a^gt2/gt2^* mice (*Figure 2B*). Taken together, these data demonstrated that SEC24A is not required for survival, development, or fertility in the mouse.

### SEC24A-deficient mice are hypocholesterolemic

SDS-PAGE analysis of non-reduced plasma samples from *Sec24a^gt/gt^* mice and their wild type littermates demonstrated no changes in several abundant plasma proteins, including albumin and transferrin

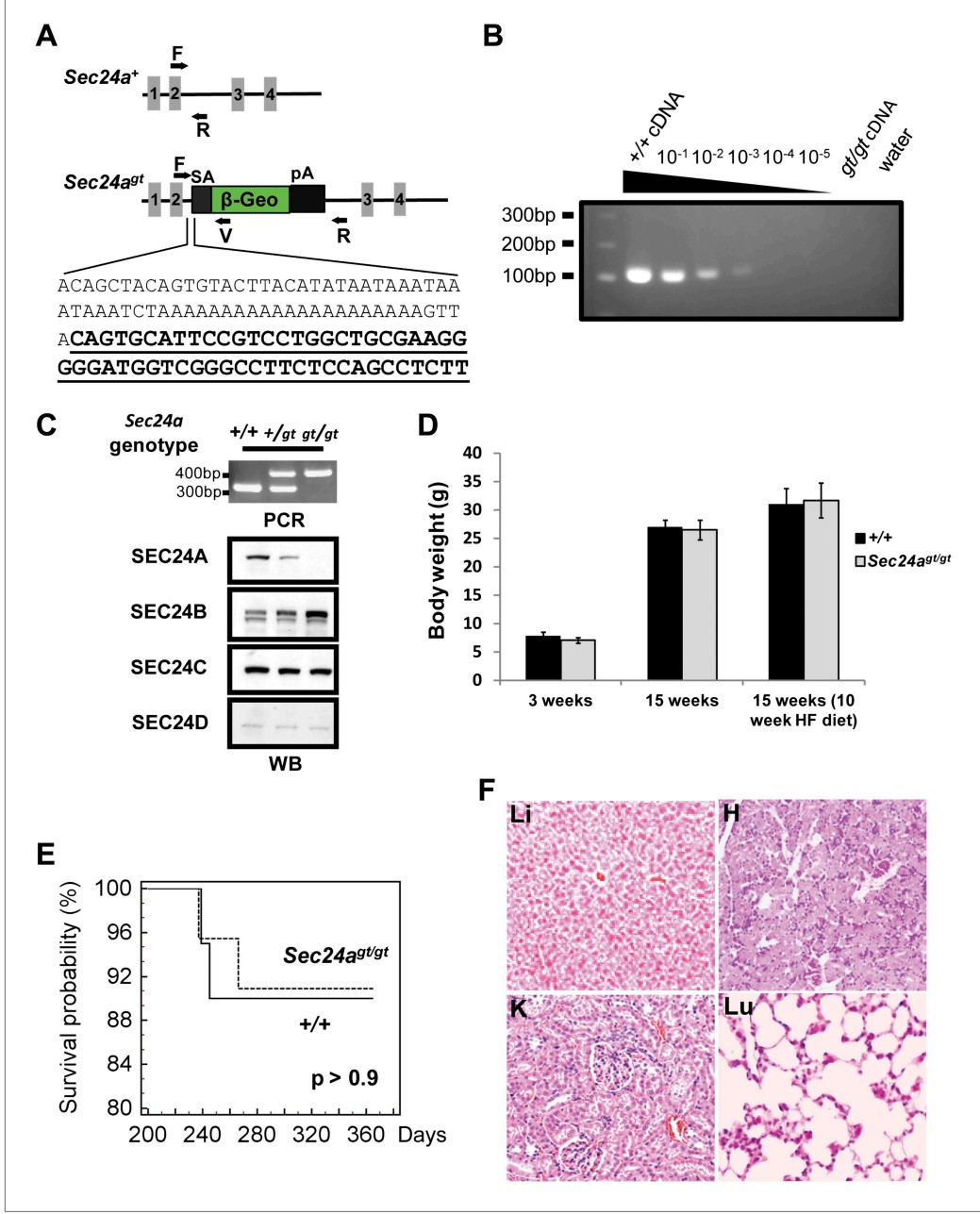

**Figure 1**. SEC24A null mice are viable and exhibit normal survival and development. (**A**) Schematic of the first *Sec24a* mutant allele (*Sec24a^gt^*). Gray blocks represent exons with specific numbers indicated. SA, splice acceptor cassette; β-Geo, β-galactosidase-*neo* fusion; pA, poly-adenylation sequence. F, R, and V represent genotyping primers. Bottom, sequence of *Sec24a^gt^* gene trap insertion junction; sequence of the gene trap cassette is underlined. (**B**) RT-PCR detection of splicing between exons 2 and 3 in *Sec24a^gt/gt^* mice. Liver cDNA of wild type mice was serially diluted into liver cDNA of *Sec24a^gt/gt^* mice as indicated and used as template for PCR with primers *Sec24a*-Exon2 and *Sec24a*-Exon3 (see primer sequences). (**C**) Loss of SEC24A protein in *Sec24a^gt/gt^* mice. Upper panel, PCR genotyping; lower panel, immunoblotting of brain protein extracts from mice with the genotypes indicated at the top, using the indicated SEC24A-D antibodies. (**D**) Body weights of SEC24A-deficient and wild type control mice. HF, high fat diet. Error bars represent SEM (standard error of the mean). At least six mice were included in each group at each time point. (**E**) Kaplan Meier plot for survival of SEC24A-deficient mice (n = 20) and littermate controls (n = 15). (**F**) Histology of several tissues from *Sec24a^gt/gt^* mice. Li, liver; H, heart; K, kidney; Lu, lung.

**Table 1.** Distributions of offspring from intercross

| Crosses | Genotype distribution in % | | | | p value (χ²) |
|---|---|---|---|---|---|
| | +/+ | +/− | −/− | | |
| Expected % | 25% | 50% | 25% | | |
| Sec24a^{+/gt} X Sec24a^{+/gt} | 25% (36) | 48.6% (70) | 26.4% (38) | | > 0.9 |
| Sec24a^{+/cgt} X Sec24a^{+/cgt} | 23.3% (7) | 43.3% (13) | 33.3% (10) | | > 0.5 |
| Sec24a^{+/gt2} X Sec24a^{+/gt2} | 27.9% (17) | 49.2% (30) | 23% (14) | | > 0.8 |
| Sec24a^{+/gt}Sec24b^{+/-} X Sec24a^{gt/gt} | Sec24a^{+/gt} 25% | Sec24a^{+/gt}Sec24b^{+/-} 25% | Sec24a^{gt/gt} 25% | Sec24a^{gt/gt}Sec24b^{+/-} 25% | |
| Observed | 20.7% (19) | 32.6% (30) | 25% (23) | 21.7% (20) | > 0.35 |
| Sec24a^{+/gt}Sec24d^{+/gt} X Sec24a^{gt/gt} | Sec24a^{+/gt} 25% | Sec24a^{+/gt}Sec24d^{+/gt} 25% | Sec24a^{gt/gt} 25% | Sec24a^{gt/gt}Sec24d^{+/gt} 25% | |
| Observed | 22.1% (21) | 29.5% (28) | 27.4% (26) | 20.1% (20) | > 0.6 |

Observed numbers are listed in parentheses.

**Table 2.** Complete blood count survey

| | WT (n = 10) | Sec24a^{gt/gt} (n = 6) | p value |
|---|---|---|---|
| WBC (X10³) | 11.77 ± 2.67 | 11.53 ± 1.60 | 0.85 |
| RBC (X10⁶) | 9.89 ± 0.41 | 9.28 ± 0.46 | 0.02 * |
| HGB (g/dl) | 12.9 ± 0.7 | 12.7 ± 0.5 | 0.51 |
| HCT (%) | 50.2 ± 1.5 | 49.2 ± 2.7 | 0.34 |
| MCV (fl) | 50.78 ± 0.97 | 52.72 ± 0.66 | 0.001 * |
| MCH (pg) | 13.06 ±0.35 | 13.48 ± 0.23 | 0.02 * |
| MCHC (%) | 25.75 ± 0.55 | 25.60 ± 0.28 | 0.55 |
| CHCM (g/dl) | 26.39 ± 0.21 | 26.28 ± 0.17 | 0.31 |
| RDW (%) | 12.92 ± 0.61 | 12.62 ± 0.70 | 0.38 |
| HDW (%) | 1.692 ± 0.086 | 1.630 ± 0.071 | 0.16 |
| PLT (X10⁴) | 123.6 ± 19.1 | 118.8 ± 8.5 | 0.58 |
| MPV (fl) | 5.2 ± 1.5 | 5.73 ± 0.2 | 0.46 |

*$p < 0.05$ by Student's t-test.

(*Figure 3A* asterisks). However, a protein migrating at ~25 kD consistently appeared under-represented by ~50% in plasma samples from *Sec24a^{gt/gt}* mice (*Figure 3A* arrow, p25). Mass spectrometry identified this protein as apolipoprotein A-I (APO-A1) (*Figure 3B*), and quantification by spectral counts or mean peak area confirmed an ~30–70% reduction of APO-A1 in *Sec24a^{gt/gt}* plasma compared to wild type. Immunoblotting confirmed an ~40% decrease in APO-A1 levels in *Sec24a^{gt/gt}* plasma (*Figure 3C*).

APO-A1 is a core protein component of high-density cholesterol-containing lipoprotein particles and is also found in some other lipoprotein particle species in the circulation (*Hoofnagle and Heinecke, 2009*). For this reason, pooled plasma samples from overnight-fasted adult male *Sec24a^{gt/gt}* mice and their wild type littermates were fractionated by FPLC and cholesterol levels in each fraction were measured (*Figure 3D*). Total cholesterol and High Density Lipoprotein (HDL) cholesterol were reduced by ~40% in *Sec24a^{gt/gt}* plasma compared to wild type, and Low Density Lipoprotein (LDL) cholesterol by ~60%. Similarly reduced plasma cholesterol levels were observed in both male and female SEC24A-deficient mice at different ages (*Figure 3E,F*). The low cholesterol phenotype was also confirmed in mice harboring the second SEC24A-deficient allele, *Sec24a^{gt2/gt2}* (*Figure 3G*).

## Loss of hepatic SEC24A expression leads to hypocholesterolemia

To further exclude a potential contribution from a passenger gene to the hypocholesterolemia phenotype (*Westrick et al., 2010*), we removed the conditional gene trap cassette from the *Sec24a^{cgt}* allele

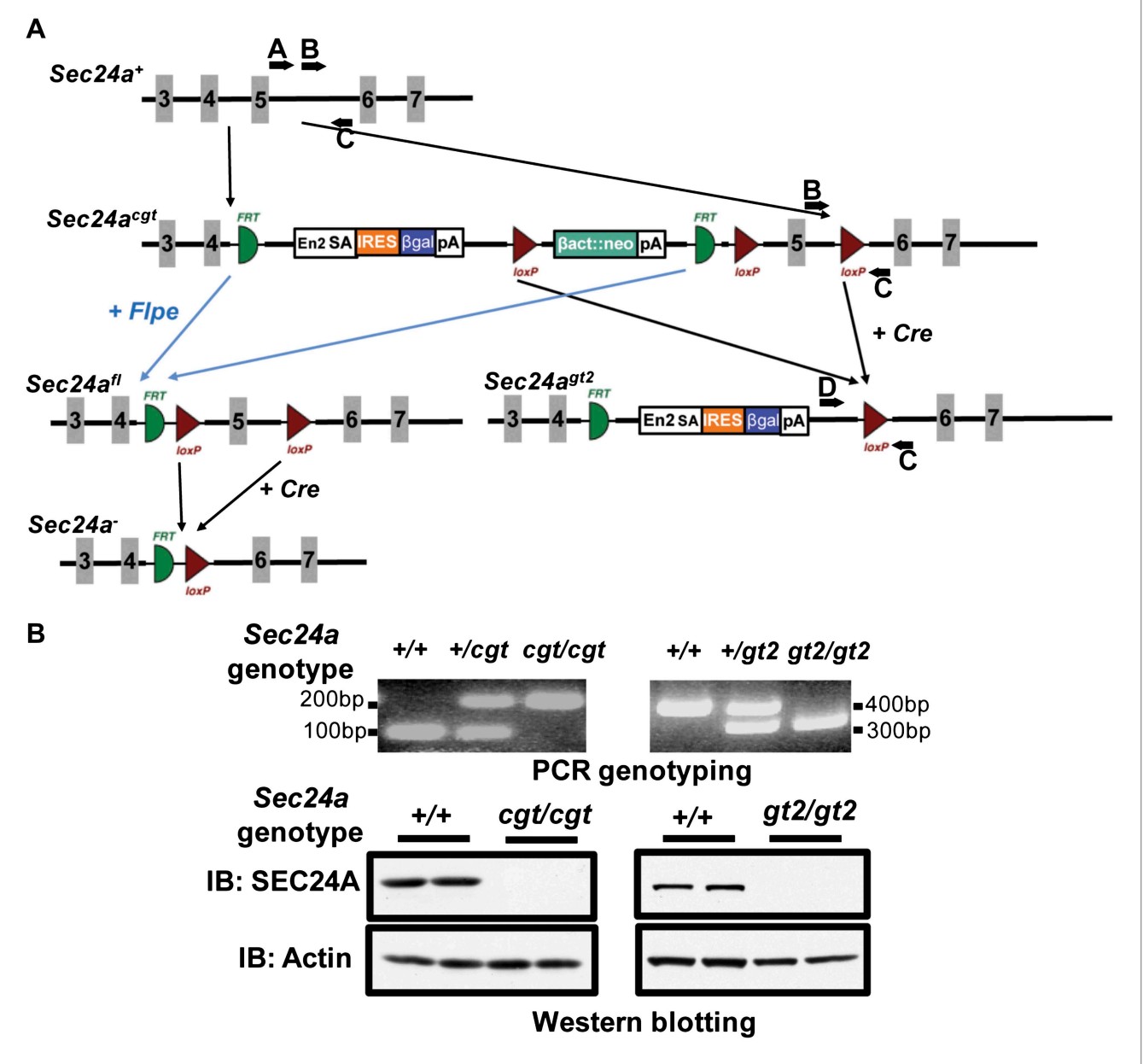

**Figure 2**. Additional targeted alleles of *Sec24a*. (**A**) Schematic of additional *Sec24a* alleles, *Sec24a^cgt^*, *Sec24a^gt2^*, *Sec24a^fl^* and *Sec24a^-^* (adapted from the Knockout Mouse Project; general conditional gene targeting scheme: https://www.komp.org/alleles.php#conditional-promoter-csd, Sec24a targeting: http://www.knockoutmouse.org/martsearch/project/24915). Gray blocks represent exons. A, B, C, and D, genotyping primers. (**B**) PCR genotyping and immuno-blot analysis of brain extracts in tissues from *Sec24a^cgt/cgt^* and *Sec24a^gt2/gt2^* mice. IB indicates immunoblotting antibody.

by germline *Flpe*-mediated excision (see Experimental procedures) to generate the *Sec24a^fl^* allele. *Sec24a^fl/fl^* mice exhibited full restoration of SEC24A expression in liver lysates (**Figure 4A**), as well as normalization of total plasma cholesterol levels (**Figure 4B**).

Analysis of public mRNA expression data revealed that *Sec24a* is ubiquitously expressed in multiple tissues (**Rosenbloom et al., 2012**). Immunoblotting confirmed that SEC24A was present in all tissues tested, although it was relatively enriched in the liver (**Figure 4C**). Hepatic *Sec24a* was selectively inactivated in *Sec24a^fl/fl^* mice by injection of a *Cre* recombinase-expressing Adenovirus (Adv-Cre). Expression of SEC24A was reduced by 70–80% in the liver of Adv-Cre-treated *Sec24a^fl/fl^* mice (**Figure 4D**), resulting in an ~25% decrease in total cholesterol levels compared to wild type control mice receiving

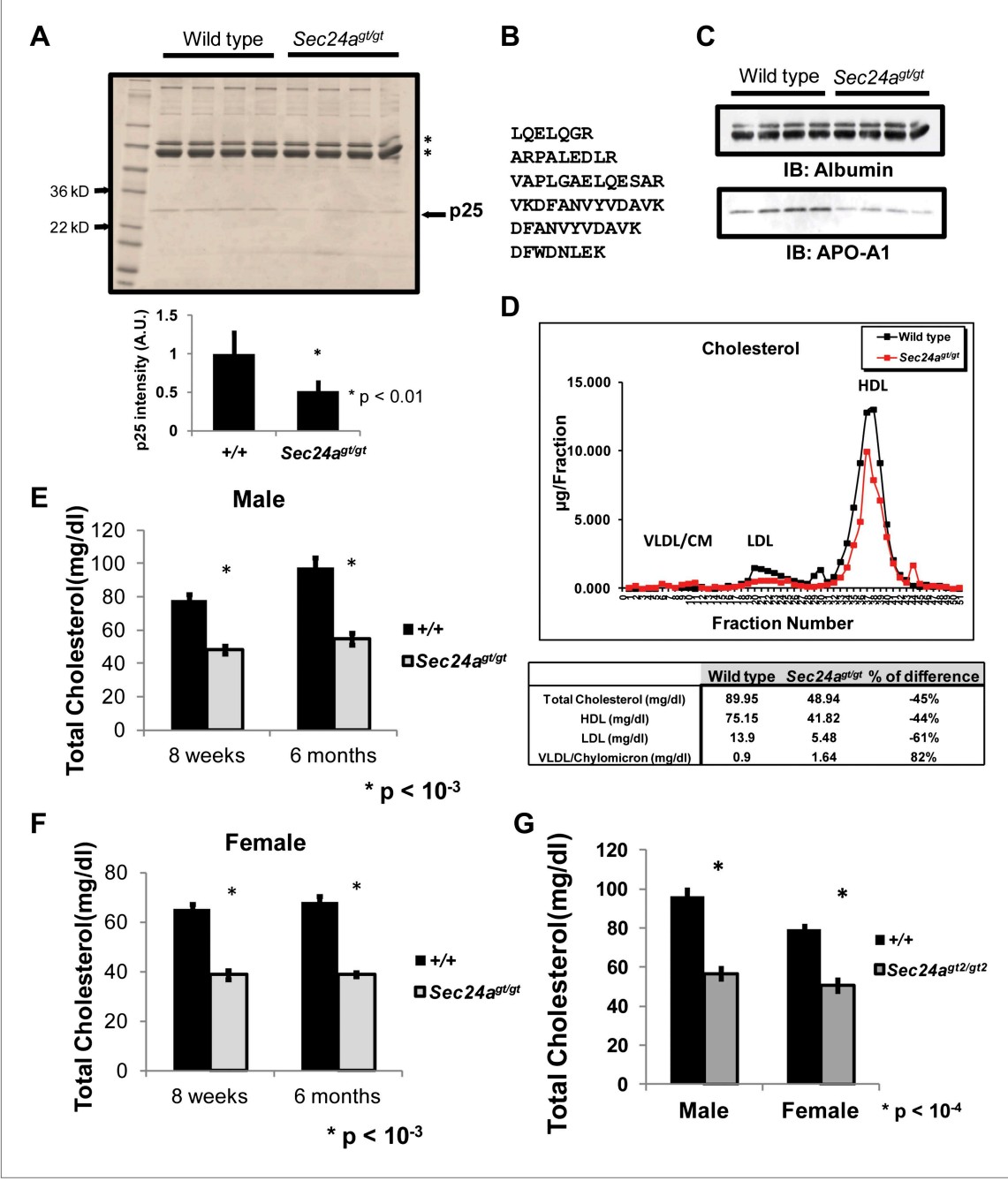

**Figure 3**. SEC24A-deficient mice develop hypocholesterolemia. (**A**) Non-reduced plasma protein samples from four wild type and four *Sec24a^gt/gt* mice were separated on SDS-PAGE and stained with coomassie brilliant blue. The first lane contains size markers. Asterisks indicate transferrin and albumin. An ~25kD protein (p25) is under-represented in the plasma of *Sec24a^gt/gt* mice. Lower panel, quantification of the intensity for the band labeled 'p25' in the upper panel; error bars represent SEM. Asterisk, p<0.01 by Student's t-test. (**B**) Identification of p25 as APO-A1 by mass spectrometry. Peptide sequences detected in HPLC-ESI-MS/MS analysis; all six peptides exhibit 100% match with mouse APO-A1 sequence. (**C**) Non-reduced plasma protein samples from four wild type or four SEC24A-deficient mice were analyzed by immunoblotting with antibodies to albumin or APO-A1. (**D**) Pooled plasma samples from seven wild type and eight *Sec24a^gt/gt* mice were fractionated by FPLC and cholesterol in each fraction quantified with a colorimetric assay; total cholesterol for the fractions containing HDL, LDL, and VLDL/Chylomicrons are indicated in the table at the bottom. (**E**) Total plasma cholesterol in male wild type control (n = 10 for 8 weeks of age, n = 4 for 6 months of age) and *Sec24a^gt/gt* (n = 5 for 8 weeks of age, n = 6 for 6 months of age). Error bars represent SEM. Asterisk, p<0.001 by Student's t-test. (**F**) Total plasma cholesterol in female wild type control (n = 7 for 8 weeks of age, n = 4 for 6 months of age) and *Sec24a^gt/gt* (n = 5 for 8 weeks of age, n = 5 for 6 months of age). Error bars represent SEM. Asterisk, p<0.001 by Student's t-test. (**G**) Total plasma cholesterol in wild type control (n = 7 for male, n = 5 for female) and *Sec24a^gt2/gt2* (n = 4 for male, n = 4 for female). Error bars represent SEM. *p<0.0001 by Student's t-test.

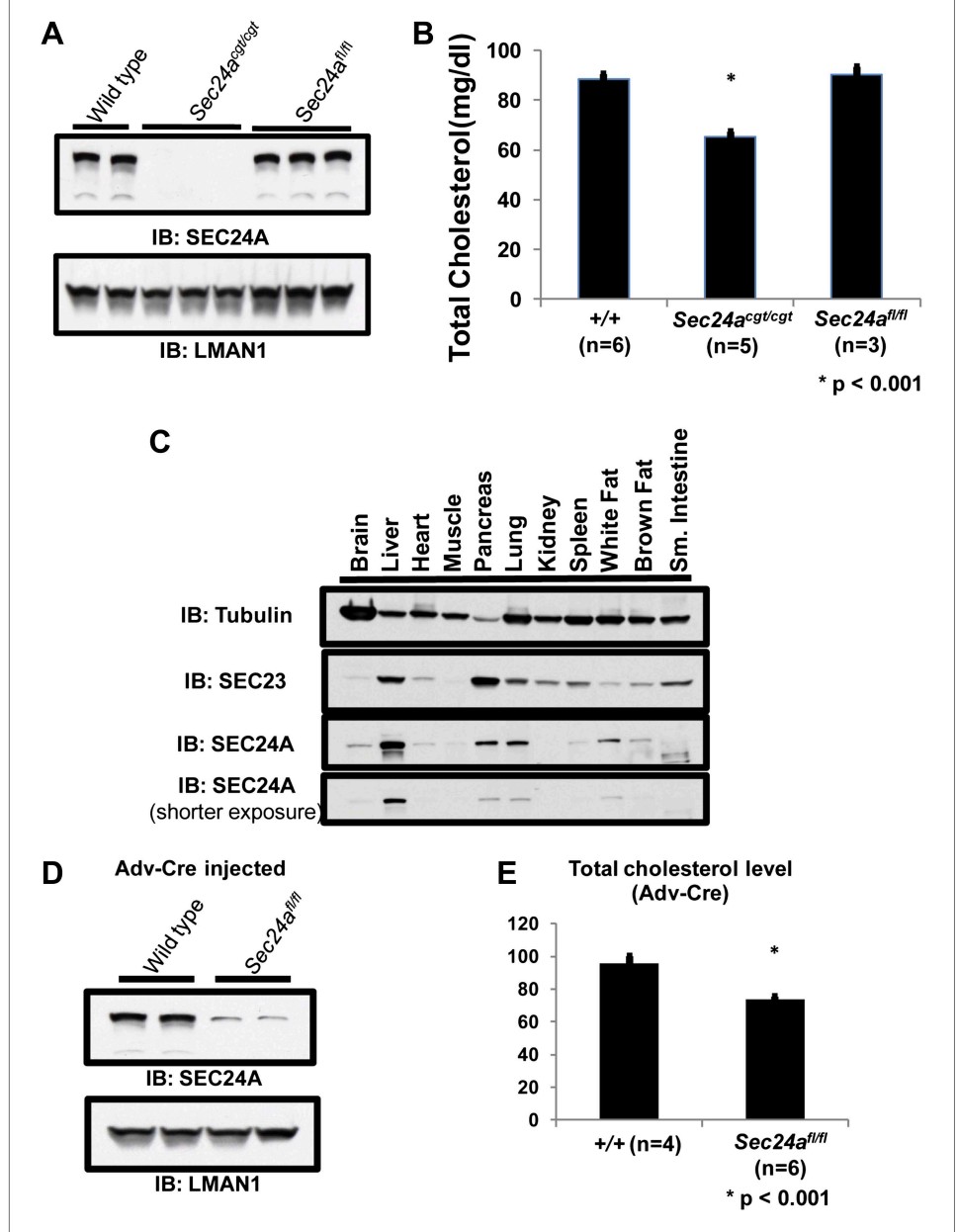

**Figure 4**. Loss of hepatic SEC24A expression leads to hypocholesterolemia. (**A**) Immunoblotting of liver lysates from wild-type and *Sec24a^cgt/cgt* mice with an anti-SEC24A and control anti-LMAN1 antibody demonstrates loss of SEC24A expression in *Sec24a^cgt/cgt* mice. Removal of the gene trap from the *Sec24a^cgt* allele to generate *Sec24a^fl* restores wild type SEC24A expression. (**B**) Total plasma cholesterol levels from wild type, *Sec24a^cgt/cgt*, and *Sec24a^fl/fl* mice were quantified with a colorimetric assay; error bars represent SEM. Asterisk, p<0.001 by Student's t-test. (**C**) Protein extracts from the indicated tissues of a wild type mouse were subjected to immunoblotting with the indicated antibodies. (**D**) Hepatic inactivation of SEC24A was performed by intravenous injection of *Sec24a^fl/fl* mice with an adenovirus encoding *Cre* recombinase (Adv-Cre). Liver protein extracts from these mice and control Adv-Cre-injected wild type mice were subjected to immunoblotting with antibodies to SEC24A or a control (LMAN1). Hepatic SEC24A is reduced by 70–80% in the Adv-Cre-treated *Sec24a^fl/fl* mice. (**E**) Total plasma cholesterol was quantified in Adv-Cre-treated mice; error bars represent SEM. n = 4 for wild type mice; n = 6 for *Sec24a^fl/fl* mice. *p<0.001 by Student's t-test.

the same adenovirus (*Figure 4E*). Although this reduction in plasma cholesterol was less than observed in *Sec24a$^{gt/gt}$* mice, considering the incomplete deletion of hepatic *Sec24a* by Adv-Cre, these data suggest that loss of hepatic *Sec24a* expression is sufficient to explain the hypocholesterolemia observed in SEC24A-deficient mice.

## SREBP signaling is unaltered by SEC24A deficiency

The SREBP1/2 transcription factors, key determinants of cholesterol metabolism (*Brown and Goldstein, 1997*, *2009*), undergo cleavage-dependent activation at the Golgi apparatus regulated by COPII-mediated transport from the ER through the chaperone protein SCAP (*Sun et al., 2005*, *2007*; *Brown and Goldstein, 2009*). Thus, specific dependence of SREBP/SCAP on SEC24A for ER-Golgi transport could potentially account for the aberrant cholesterol metabolism in SEC24A-deficient mice. To test this hypothesis, we performed mRNA-Seq (*Mortazavi et al., 2008*) to examine the hepatic transcriptome in *Sec24a$^{gt/gt}$* mice (*Figure 5A*, *Supplementary file 1*). No significant differences were observed in the expression levels of the 20 most abundant hepatic transcripts between wild type and *Sec24a$^{gt/gt}$* mice (<1.2 fold, False Discovery Rate [FDR] > 0.4, *Figure 5B*). Among the ~20,000 identified transcripts, only ~30 showed significant down-regulation (>2 fold, FDR < 0.05, RPKM > 0.2), with *Sec24a* as the most significantly down-regulated mRNA in *Sec24a$^{gt/gt}$* liver (*Figure 5C*). Gene-ontology enrichment analysis failed to identify a consistent pattern in the changes in transcript levels, though modest shifts were noted in a few metabolic genes (*Supplementary file 2*). There was no significant correlation between the gene expression profile for *Sec24a$^{gt/gt}$* mice and previously reported profiles from mice with altered expression of SCAP or SREBP1/2 (*Horton et al., 2003*). Specifically, transcripts for direct SREBP1/2 targets, including the Low Density Lipoprotein Receptor (LDLR), HMG-CoA reductase, and PCSK9, remained unchanged by mRNA-seq and qPCR, though SCD1 was decreased by ~50% in *Sec24a$^{gt/gt}$* mice (*Figure 5D*), as observed by mRNA-seq (FDR < 3 × 10$^{-13}$). In addition, no detectable difference in the extent of SREBP1/2 cleavage was observed by immunoblotting of liver lysates from *Sec24a$^{gt/gt}$* mice and their wild type littermates (*Figure 5E*).

Of note, RNA-seq detected expression of all four SEC24 paralogs in normal liver, with the number of transcripts for *Sec24c* and *Sec24d*–twofold greater than *Sec24a* and 5–10 fold greater than *Sec24b* (*Figure 5F*). Quantitative immunoblotting using RFP-tagged SEC24A and SEC24C/D as references confirmed the presence of SEC24C/D proteins at similar levels to SEC24A in wild type liver, and no change in SEC24C/D levels in *Sec24a$^{gt/gt}$* liver (*Figure 5G*).

## SEC24A deficiency up-regulates LDLR protein levels by decreasing circulating PCSK9

The reduction in circulating cholesterol in *Sec24a$^{gt/gt}$* mice could result from decreased hepatic output or increased clearance by the liver. To distinguish these two mechanisms, we crossed *Sec24a$^{gt/gt}$* mice with *Apoe* null mice, in which receptor-mediated clearance of cholesterol-rich lipoprotein remnant particles from the plasma is inhibited due to loss of the critical lipoprotein particle component APOE. Consistent with previous reports (*Piedrahita et al., 1992*; *Zhang et al., 1992*), *Apoe$^{-/-}$* mice showed markedly elevated cholesterol levels (272 ± 13 mg/dl, n = 8). However, in contrast to the ~45% reduction in plasma cholesterol observed in mice singly deficient for *Sec24a* (*Figure 3D*), the plasma cholesterol levels in *Apoe$^{-/-}$Sec24a$^{gt/gt}$* mice (277 ± 14 mg/dl, n = 7) were indistinguishable (p>0.8) from singly *Apoe$^{-/-}$* littermates (*Figure 6A*). Thus, *Apoe* is epistatic to *Sec24a*, suggesting that SEC24A may regulate cholesterol metabolism via the receptor-mediated clearance pathway. To further test this hypothesis, we also crossed *Sec24a$^{gt/gt}$* mice into the *Ldlr* null background. Consistent with previous reports (*Ishibashi et al., 1993*), *Ldlr$^{-/-}$* mice exhibited markedly elevated plasma cholesterol levels (211 ± 9 mg/dl, n = 14). Doubly deficient *Ldlr$^{-/-}$Sec24a$^{gt/gt}$* mice also displayed similarly elevated plasma cholesterol levels (197 ± 12 mg/dl, n = 11), indistinguishable (p>0.35) from their *Ldlr$^{-/-}$* littermates (*Figure 6B*). Taken together, these data suggest that reduction in plasma cholesterol in SEC24A-deficient mice could result from increased clearance of cholesterol-rich lipoproteins by the LDLR. Consistent with this hypothesis, immunoblotting demonstrated elevated LDLR levels in liver protein extracts or in proteins bound to a wheat germ agglutinin (WGA) matrix from *Sec24a$^{gt/gt}$* mice compared to littermate controls, whereas other proteins including APOE, APOB48/100, or APO-A1 showed little alteration in the liver (*Figure 6C*). Since hepatic *Ldlr* transcripts remained unaltered in *Sec24a$^{gt/gt}$* by both mRNA-Seq and qPCR (*Figure 5D*), these data indicate that the reduction in LDLR protein levels in SEC24A-deficient mice was mediated via a translational or post-translational mechanism.

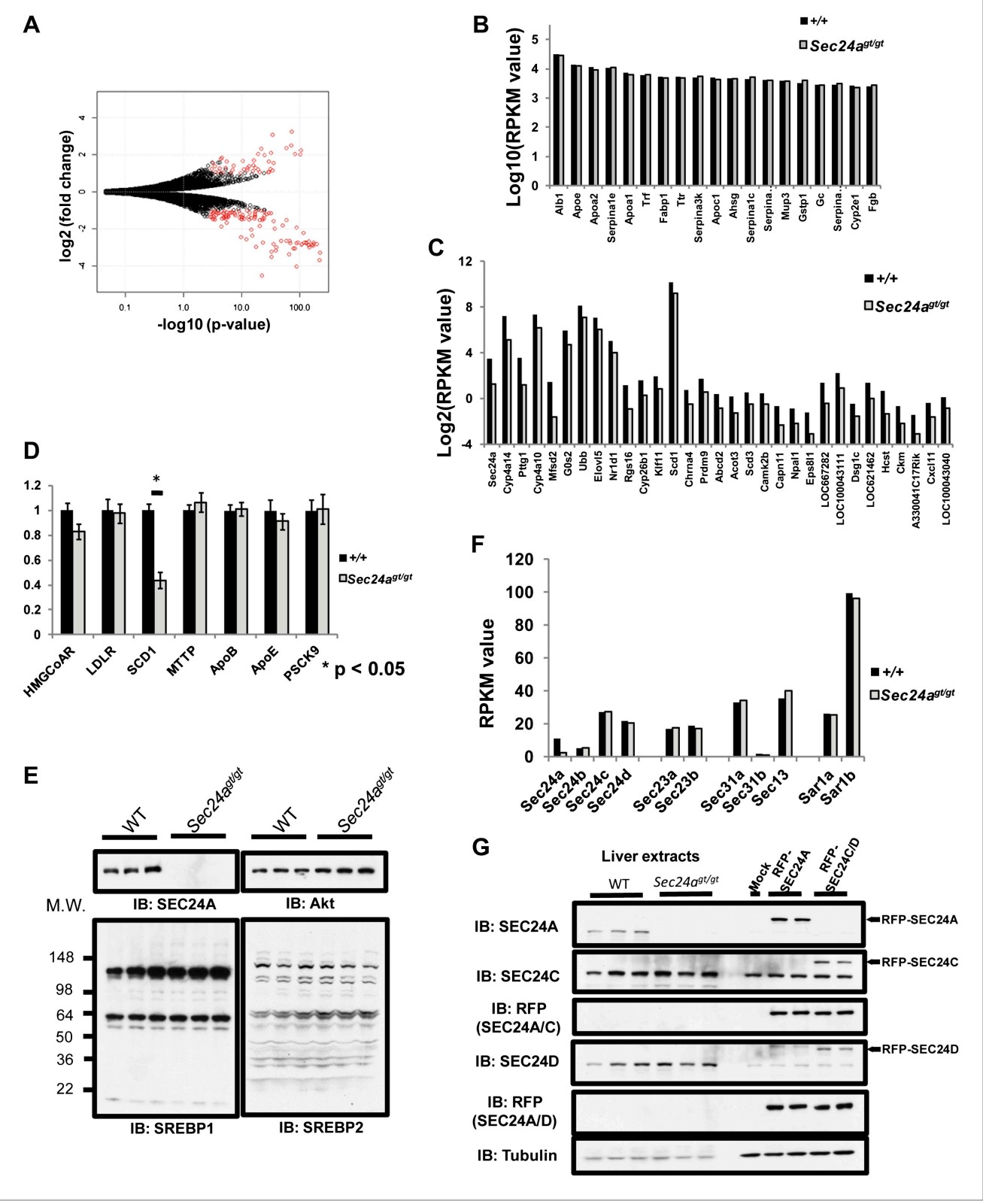

**Figure 5**. SEC24A deficiency does not alter SREBP signaling. (**A**) Volcano plot of liver transcriptome analysis by mRNA-Seq. X-axis, -Log(p-value); y-axis, $\text{Log}_2$(fold difference WT/*Sec24a*$^{gt/gt}$). Significantly altered genes (fold change > 2, RPKM > 0.1, and FDR < 0.05) are colored in red. (**B**) Twenty most abundant hepatic transcripts detected in wild type and *Sec24a*$^{gt/gt}$ mice by mRNA-Seq. X-axis, gene name; y-axis: Log(RPKM value). (**C**) Hepatic transcripts significantly down-regulated by SEC24A deficiency. X-axis, gene name; y-axis: $\text{Log}_2$(RPKM value). (**D**) Liver mRNA samples from wild type (n = 4) or
*Figure 5. Continued on next page*

*Figure 5. Continued*

SEC24A-deficient mice (n = 4) were subjected to quantitative-PCR (q-PCR) with primers for the indicated SREBP-regulated transcripts. Error bars represent SEM. Asterisk, p<0.01 by Student's t-test. (**E**) Liver protein extracts from wild type and *Sec24a^gt/gt* mice were subjected to immunoblotting with the indicated antibodies. (**F**) Transcript abundance detected by mRNA-Seq for COPII genes in the liver of wild type and *Sec24a^gt/gt* mice. (**G**) SEC24A and SEC24C/D are expressed at comparable levels in the liver. Liver protein extracts from three wild type and three *Sec24a^gt/gt* mice and 293T cells expressing RFP-tagged SEC24A, SEC24C or SEC24D as references were analyzed with the indicated antibodies.

LDLR levels are regulated by intracellular factors such as the E3 ligase IDOL (*Zelcer et al., 2009*) and circulating PCSK9. The latter is secreted into the plasma primarily by hepatocytes, binding to LDLR and promoting its endocytosis and degradation (*Costet et al., 2008*; *Horton et al., 2009*). No difference in the level of IDOL was observed in immunoblotting of liver lysates prepared from SEC24A-deficient mice compared to their wild type littermates (*Figure 6C*). To assess a potential role for PCSK9 in the low plasma cholesterol phenotype of SEC24A-deficient mice, we measured circulating PCSK9 levels by ELISA. An ~55% reduction of plasma PCSK9 was observed in *Sec24a^gt/gt* mice (*Figure 6D*), suggesting that SEC24A-deficiency may lower plasma cholesterol via reduced secretion of PCSK9.

## PCSK9 is a soluble COPII cargo regulated by SEC24A

No change in PCSK9 hepatic mRNA was observed between *Sec24a^gt/gt* mice and their wild type littermates by mRNA-seq or qPCR (*Figure 5D*), suggesting a translational or post-translational mechanism for the reduction of circulating PCSK9 in SEC24A-deficient mice. Analysis of liver protein extracts by immunoblotting demonstrated significant accumulation of both mature and pro-PCSK9 in the liver of *Sec24a^gt/gt* mice compared to littermate controls (*Figure 7A*). To determine the site of PCSK9 intracellular accumulation, we treated liver lysates with endoglycosidase H (EndoH). Nearly all of the intracellularly-accumulated PCSK9 in *Sec24a^gt/gt* liver exhibited sensitivity to EndoH with faster eletrophoresis mobility (*Figure 7B*), suggesting that SEC24A deficiency leads to accumulation of PCSK9 in the ER.

These data suggest that COPII-dependent ER-Golgi transport plays a critical role in PCSK9 secretion from the cell. Belfeldin A (BFA) disrupts the Golgi apparatus and consequently the delivery of ER-derived COPII vesicles along the secretory pathway. When BFA was applied to a 293T cell line stably expressing PCSK9 fused with a C-terminal FLAG tag (PCSK9-FLAG), PCSK9 secretion into the medium was abolished and the protein accumulated intracellularly (*Figure 7C*). To directly test whether the COPII machinery plays a role in ER exiting of PCSK9, we employed permeabilized 293T cells stably expressing PCSK9-FLAG in an in vitro COPII vesicle budding assay. In the presence of rat liver cytosol supplying COPII components, along with ATP and GTP, PCSK9 was packaged into COPII vesicles that sedimented in a high-speed pellet fraction, together with the classic COPII cargo SEC22B, but not the ER resident protein roboporin (*Figure 7D*). ER budding of PCSK9 and SEC22B-containing vesicles were inhibited by GTPyS as well as constitutively active SAR1 (SAR1A H79G), both of which prevent GTP hydrolysis and subsequently COPII vesicle budding from the ER membrane (*Figure 7D*). Inhibition of COPII vesicle formation in cells by SAR1B H79G also prevented secretion of PCSK9 into the medium, with corresponding intracellular accumulation of the protein (*Figure 7E*). These data demonstrate that PCSK9 secretion is dependent on ER exiting in COPII-coated vesicles.

SEC24 plays a central role in the recruitment of cargos to the COPII vesicle (*Miller et al., 2002*, *2003*), with different paralogs thought to provide specificity in cargo selection (*Mancias and Goldberg, 2008*; *Zanetti et al., 2011*). To test whether SEC24 paralogs provide specificity in transporting PCSK9, we introduced RFP-tagged SEC24A-D paired with GFP-tagged SEC23A into 293T cells stably expressing PCSK9-FLAG. SEC24A and SEC24B, but not SEC24C or SEC24D, associated with PCSK9 precipitated by anti-FLAG antibody in CHAPS-based buffer (*Figure 8A*), suggesting that SEC24A and SEC24B may selectively mediate PCSK9 ER exiting and secretion. To directly test this hypothesis, we introduced RFP-tagged SEC24A-D paired with GFP-tagged SEC23A together with PCSK9-FLAG into 293T cells. Expression of SEC24A paired with SEC23A (5.4 ± 1.7 fold compared to RFP control, p<0.05), and to a lesser extent SEC24B/SEC23A (2.1 ± 0.4 fold compared to RFP control, p<0.05), reduced intracellular levels of PCSK9 and led to increased secretion of the protein into the medium, while SEC24C or SEC24D in complex with SEC23A had little effect (*Figure 8B,C*). This effect was absent when ER-Golgi transport was blocked by BFA treatment, further suggesting that SEC24A, and to a lesser extent SEC24B, selectively mediate COPII-dependent secretion of PCSK9.

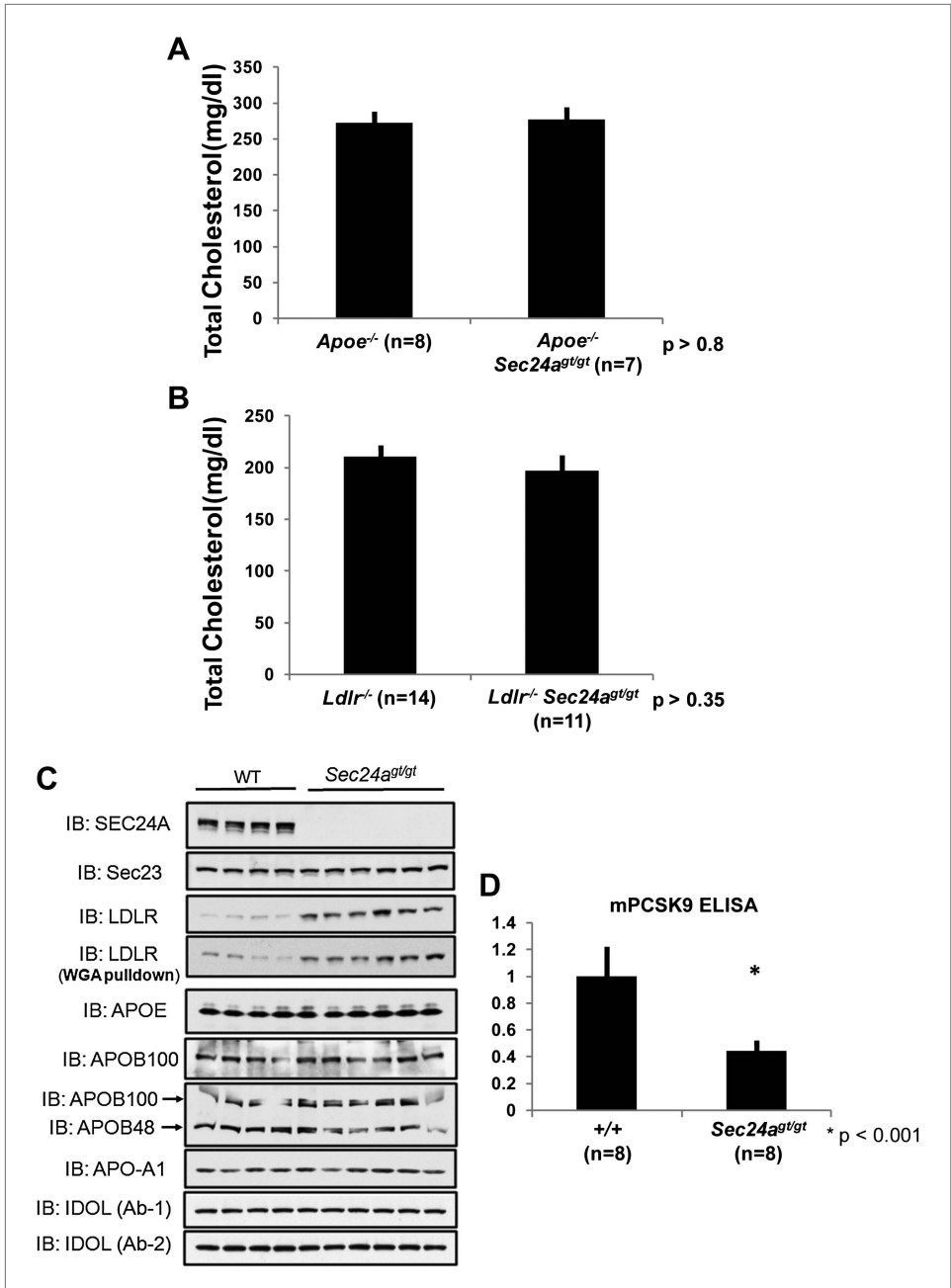

**Figure 6**. SEC24A deficiency up-regulates LDLR protein levels by decreasing circulating PCSK9. (**A**) SEC24A deficiency does not cause hypocholesterolemia in the setting of APOE deficiency. Total plasma cholesterols in *Apoe*$^{-/-}$ (n = 8) and *Apoe*$^{-/-}$*Sec24a*$^{gt/gt}$ (n = 7) mice. Error bars represent SEM. p=~0.8 by Student's t-test. (**B**). Total plasma cholesterol levels from *Ldlr*$^{-/-}$ (n = 14) and *Ldlr*$^{-/-}$*Sec24a*$^{gt/gt}$ (n = 11) mice. Error bars represent SEM. p=~0.35 by Student's t-test. (**C**) Liver protein extracts from wild type and *Sec24a*$^{gt/gt}$ mice were subjected to immunoblotting with the indicated antibodies. (**D**) Plasma PCSK9 levels from wild type (n = 8) and *Sec24a*$^{gt/gt}$ (n = 8) mice were quantified by ELISA. Error bars represent SEM. *p<0.001 by Student's t-test.

To more directly examine the dependence of PCSK9 on SEC24A for efficient exit from the ER, the in vitro COPII vesicle budding assay was performed using cytosols from rat hepatoma McA-RH777 cells treated with control siRNA, or siRNAs against SEC24A, SEC24B, or both proteins. Depletion of SEC24A/B was confirmed by immunoblotting (*Figure 8D*). Compared to control siRNA treatment,

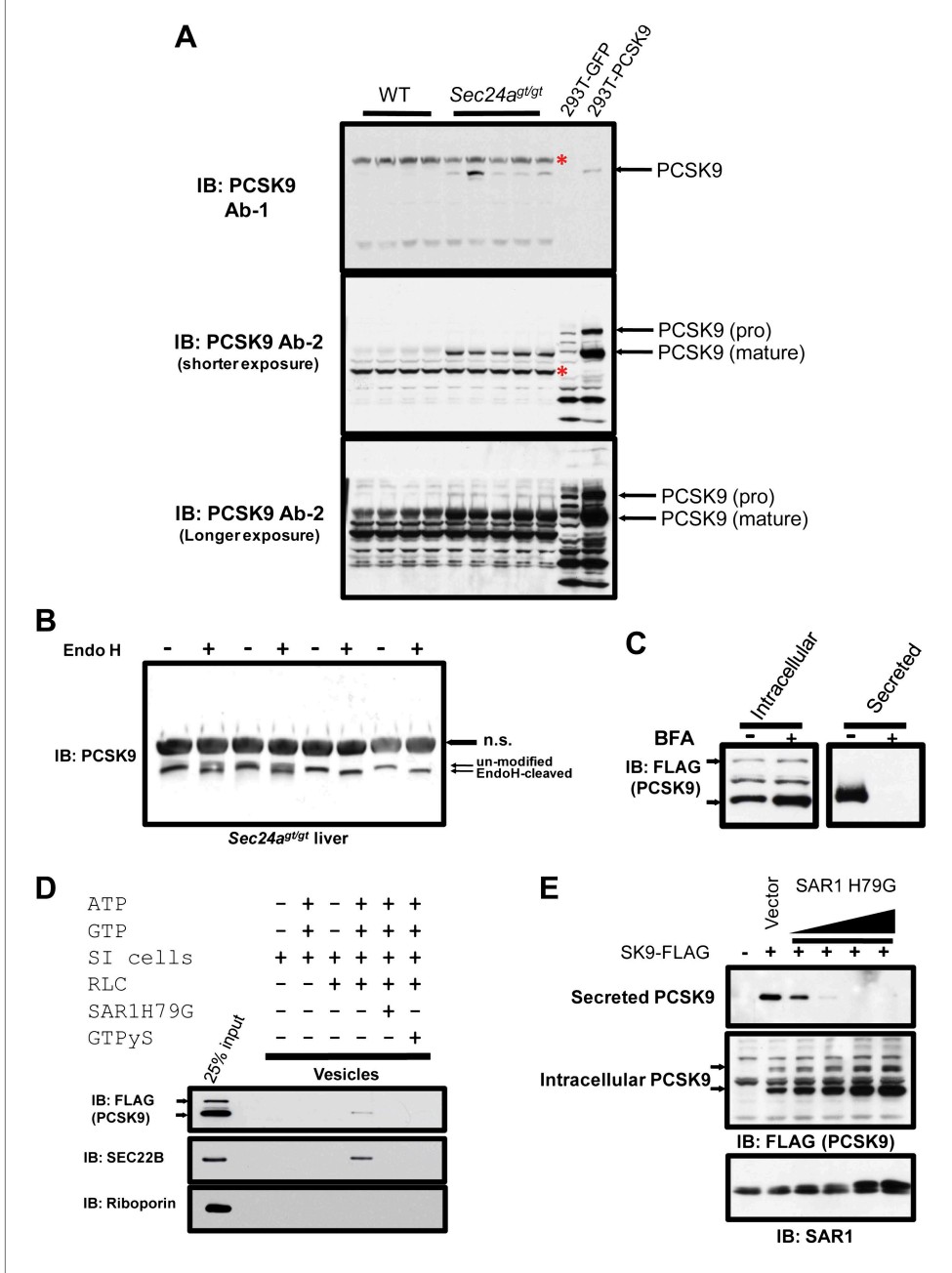

**Figure 7**. PCSK9 is a soluble COPII cargo. (**A**) Liver protein extracts from wild type and *Sec24a$^{gt/gt}$* mice and cell lysates from 293T cells expressing PCSK9 were subjected to immunoblotting with two different anti-PCSK9 antibodies. Red asterisks indicate non-specific bands. (**B**) Liver protein extracts from SEC24A-deficient mice were subjected to Endo H treatment before SDS-PAGE and immunoblotting with an anti-PCSK9 antibody. n.s., non-specific band. (**C**) Cell lysates and conditioned medium from 293T cells stably expressing PCSK9-FLAG were analyzed by immunoblotting with an anti-FLAG antibody following treatment with or without BFA. The arrows indicate un-cleaved (upper) and auto-cleaved (lower) forms of PCSK9. (**D**) Permeabilized 293T cells stably expressing PCSK9-FLAG were employed in an in vitro COPII budding assay; the resulting vesicle fractions and permeabilized cell inputs were separated by SDS-PAGE and visualized by immunoblotting. The arrows indicate un-cleaved (upper) and auto-cleaved (lower) forms of PCSK9. (**E**) Cell lysates and conditioned medium from 293T cells stably expressing PCSK9-FLAG transfected with a vector control or a plasmid expressing a dominant-negative mutant SAR1 (H79G) and analyzed by immunoblotting with an anti-FLAG antibody. The arrows indicate un-cleaved (upper) and auto-cleaved (lower) forms of PCSK9.

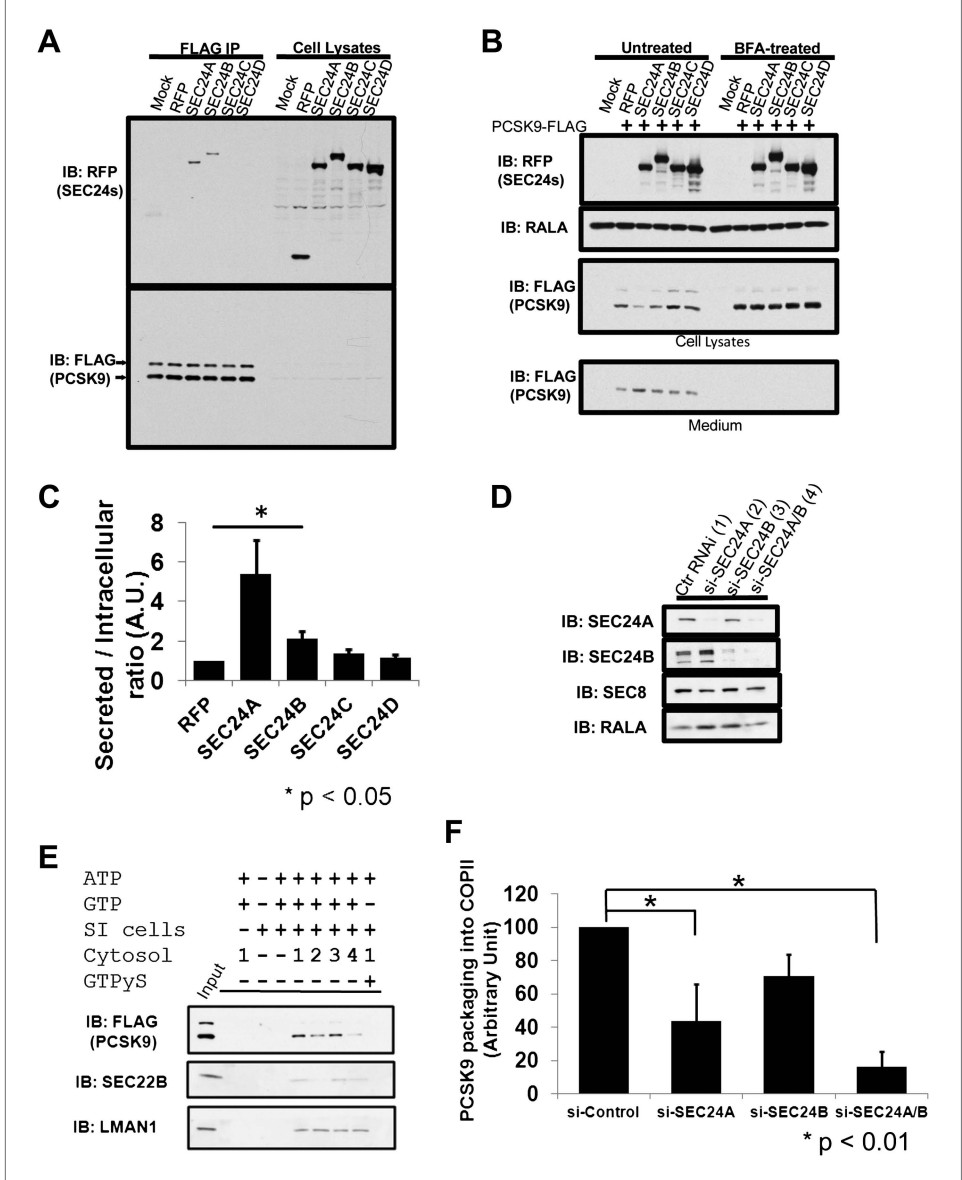

**Figure 8**. SEC24A regulates PCSK9 secretion. (**A**) Cell lysates from 293T cells stably expressing PCSK9-FLAG transfected with plasmids expressing RFP-tagged SEC24A-D or a control RFP vector together with GFP-tagged SEC23A, subjected to immune-precipitation; and immune-complexes and cell lysates then examined by immuno-blotting with anti-RFP or anti-FLAG antibodies. The arrows indicate un-cleaved (upper) and auto-cleaved (lower) forms of PCSK9. (**B**) Cell lysates and conditioned medium from 293T cells co-transfected with PCSK9-FLAG and SEC23/24 plasmids as in (**A**) were subjected to immunoblotting with the indicated antibodies. Cells treated with BFA were employed as controls. (**C**) Ratio of secreted PCSK9/intracellular PCSK9 for each transfected RFP-tagged SEC24 and control. Quantification was performed from five independent experiments. Error bars represent SEM. Asterisk, p<0.05 by Student's t-test. (**D**) Deficient McA-RH777 cells treated with the indicated siRNAs were subjected to immunoblotting to determine SEC24A and SEC24B levels. Numbers indicate the cells receiving different siRNAs. (**E**) Deficient McA-RH777 cells treated with the indicated siRNAs used as the source of cytosol for in vitro COPII budding assay as in (7d). (**F**) Quantification of PCSK9 packaging into COPII vesicles from four different experiments. Error bars represent standard deviation. * p<0.01 by Student's t-test.

PCSK9 packaging into COPII vesicles was reduced upon depletion of SEC24A (*Figure 8E,F*), and to a greater extent with depletion of both SEC24A and SEC24B. Similar reduction in COPII vesicle packing was observed for SEC22B, consistent with the previous report of its selectivity as a cargo for SEC24A/B (*Mancias and Goldberg, 2007*). In contrast, packaging of the non-selective cargo, LMAN1, was not

significantly altered by depletion of SEC24A, SEC24B, or both, consistent with previous reports (*Mancias and Goldberg, 2007*; *Wendeler et al., 2007*).

## SEC24 paralogs play divergent roles in vivo with limited overlapping function

To test the selectivity of SEC24 paralogs in vivo, we performed crosses between SEC24B- and SEC24A-deficient mice (*Table 1*). Haplo-deficiency for both *Sec24a* and *Sec24b* resulted in no discernible phenotype and normal plasma cholesterol levels (*Figure 9A*). However, *Sec24a^{gt/gt}* mice that were also haplo-deficient for *Sec24b* exhibited a further ~25% reduction in plasma cholesterol compared to *Sec24a^{gt/gt}* littermates (*Figure 9A*), although they appeared otherwise normal. These data suggest a partial overlap in function between SEC24A and SEC24B.

The possibility of genetic interaction between *Sec24a* and *Sec24d* was also examined (*Table 1*). As for SEC24B above, *Sec24a^{+/gt}Sec24d^{+/gt}* compound heterozygous mice also exhibited no discernible phenotype and had normal plasma cholesterol (*Figure 9B*). Although SEC24D deficiency in mice results in early embryonic lethality (*Baines et al., In press*), *Sec24a^{gt/gt}Sec24d^{+/gt}* mice were also viable (*Table 1*) and displayed no further reduction in plasma cholesterol compared to *Sec24a^{gt/gt}* littermates (*Figure 9B*), indicating distinct functions between SEC24D and SEC24A/B.

## Discussion

The finding that complete deficiency of the COPII subunit SEC24A is compatible with normal survival and development in the mouse is surprising, in light of its ubiquitous expression and presumed fundamental function in the secretory pathway. However, examination of these animals uncovered an unexpected reduction in plasma cholesterol due to a specific block in the secretion of PCSK9, a circulating regulator of cell surface LDL receptor (*Costet et al., 2008*; *Horton et al., 2009*). Genetic crosses demonstrate that both *Apoe* and *Ldlr* are epistatic to *Sec24a*, suggesting that SEC24A primarily affects receptor-mediated cholesterol clearance of cholesterol-rich lipoproteins. Consistent with these genetic data, hepatic LDLR levels are up-regulated in SEC24A deficient mice as a consequence of a specific dependence of PCSK9 on SEC24A for efficient exit from the ER. Our findings also identify a partial overlap in cargo selectivity between SEC24A and B, and suggest a previously unappreciated heterogeneity in the recruitment of secretory proteins to the COPII vesicles that extends to soluble as well as trans-membrane cargos.

Yeast SEC24p play a central role in the recruitment of cargo proteins into COPII vesicles to enter the secretory pathway (*Miller et al., 2002*, *2003*), and the non-essential SEC24 homologs LST1p and ISS1p may co-operate with SEC24p for efficient sorting of a specialized set of cargos (*Roberg et al., 1999*; *Peng et al., 2000*; *Shimoni et al., 2000*). The expansion of the SEC24 family in vertebrates may have facilitated the accommodation of a wider range of cargo proteins (*Jensen and Schekman, 2011*; *Zanetti et al., 2011*). This notion is supported by the diverse phenotypes observed in vertebrates with selective deficiency of a single SEC24 paralog. The mammalian subfamily of SEC24A/B exhibits greater

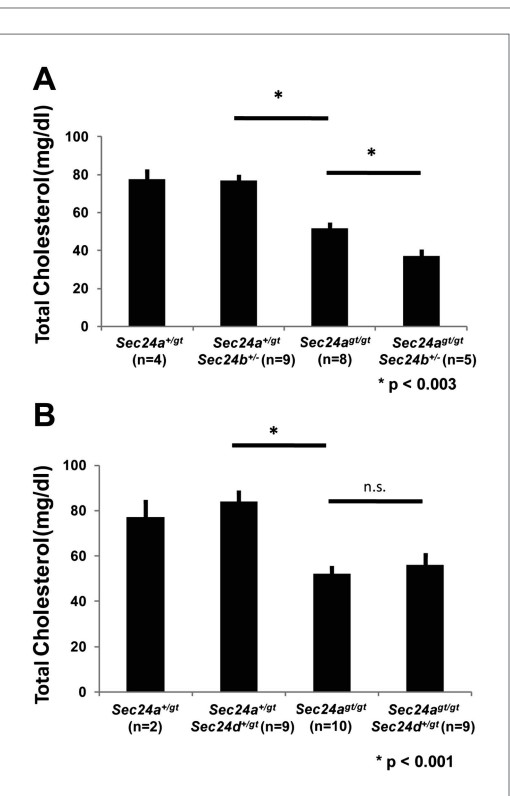

**Figure 9**. SEC24B but not SEC24D exhibit partial overlap in function with SEC24A *in vivo*. (**A**) Total plasma cholesterol levels from mice generated from a *Sec24a^{+/gt}Sec24b^{+/-}* X *Sec24a^{gt/gt}* cross. Error bars represent SEM. Asterisk, p<0.003 by Student's t-test. (**B**) Total plasma cholesterol levels from mice generated from a *Sec24a^{+/gt}Sec24d^{+/gt}* X *Sec24a^{gt/gt}* cross. Error bars represent SEM. * p<0.003; n.s., p=~0.6, by Student's t-test.

sequence identity than SEC24C/D to the essential yeast gene product SEC24p, with SEC24C/D closer to the non essential yeast gene products LST1p and ISS1p (*Peng et al., 2000*; *Mancias and Goldberg, 2008*; *Zanetti et al., 2011*). However, SEC24A-deficient mice are remarkably normal, aside from reduced plasma cholesterol. Even the more severe neural tube closure defect in SEC24B-deficient mice (*Merte et al., 2010*) stands in stark contrast to the very early embryonic lethality in SEC24D-deficient mice (*Baines et al., In press*).

This wide range of SEC24-deficient phenotypes in vertebrates could be explained by either (1) paralog-specific sorting selectivity encoded by intrinsic cargo selectivity or (2) tissue/cell-type-specific differences in relative abundance. Our data, together with those of Merte et al. (*Merte et al., 2010*), suggest that intrinsic cargo selectivity among SEC24 paralogs is a key determinant of SEC24 function. The latter authors demonstrated specific dependence of VANGL2, a key component of the planar cell polarity signaling pathway, on the SEC24B paralog for ER exiting (*Merte et al., 2010*). Our findings identify PCSK9 as a specific SEC24A-dependent COPII cargo. Although all four SEC24 paralogs are expressed in the liver, loss of SEC24A disrupts PCSK9 secretion without affecting other COPII-dependent processes such as SREBP activation. SEC24B exhibits limited compensation for SEC24A, with combined genetic deficiencies providing no evidence for functional overlap with SEC24D. These data suggest overlapping cargo specificities between the members of the SEC24A/B subfamily, yet distinct from SEC24C/D. Furthermore, steady state levels of SEC24B, but not SEC24C/D, are increased in the absence of SEC24A (*Figure 1C*), although SEC24B mRNA levels remain constant. These data imply that SEC24A/B are balanced in a cytoplasmic pool distinct from SEC24C/D. Taken together, our data raise the possibility of extensive heterogeneity among COPII vesicles and their function in cargo selection and export, determined at least in part by SEC24 paralog composition.

Intrinsic selectivity of cargos by SEC24s presumably relies on specific protein-protein interactions (*Lee et al., 2004*; *Zanetti et al., 2011*). Trans-membrane proteins including SEC22, Syntaxin 5, APP1, and VANGL2, could preferentially interact with some but not all SEC24 paralogs for differential recruitment to the COPII vesicle (*Mossessova et al., 2003*; *Kim et al., 2007*; *Mancias and Goldberg, 2008*; *Merte et al., 2010*). In contrast, the cargo receptor LMAN1 interacts equally with all four SEC24 paralogs (*Mancias and Goldberg, 2007*; *Wendeler et al., 2007*). Structural studies have revealed distinct cargo binding sites among SEC24 paralogs that recognize different sorting signals in specific trans-membrane cargos (*Mancias and Goldberg, 2008*). Nevertheless, whether and how soluble cargos are selectively transported via COPII vesicles remain to be fully defined (*Warren and Mellman, 1999*; *Lee et al., 2004*; *Jensen and Schekman, 2011*).

The 'bulk flow' model proposes that soluble cargos exit the ER by default without selection (*Wieland et al., 1987*; *Martinez-Menarguez et al., 1999*; *Warren and Mellman, 1999*; *Thor et al., 2009*), and would therefore predict that alterations of overall SEC24 abundance would proportionally affect secretion of all soluble cargos. The selective dependence of PCSK9 on SEC24A (and to a lesser extent on SEC24B), is inconsistent with this model. Indeed, PCSK9 represents the first example of a soluble vertebrate cargo that is differentially regulated by specific interaction with selective components of the COPII machinery. This specificity might be conferred by a cargo receptor that recruits PCSK9 into COPII vesicles for efficient secretion, in contrast to the bulk flow of more abundant soluble secretory proteins (*Martinez-Menarguez et al., 1999*; *Warren and Mellman, 1999*).

Yeast α-factor is recruited to COPII vesicles and interacts with SEC24 (*Kuehn et al., 1998*; *Shimoni et al., 2000*), likely through the cargo receptor ERV29p (*Malkus et al., 2002*). Genetic and biochemical studies have identified the LMAN1/MCFD2 complex as the cargo receptor for the soluble coagulation factors V and VIII in mammals (*Nichols et al., 1998*; *Zhang et al., 2003*, *2005*; *Khoriaty et al., 2012*). Additionally, TANGO1 localized to ER exit sites facilitates loading of bulky cargos such as collagen VII (*Saito et al., 2009*, *2011*). The existence of a PCSK9 ER cargo receptor has been proposed (*Nassoury et al., 2007*), but its identity has been unclear. Alternatively, COPII vesicles coated by SEC24C/D might exclude selected soluble cargos such as PCSK9. Additional cytosolic factor may also contribute to the selectivity among SEC24 paralogs as observed in the case of VANGL2 sorting (*Merte et al., 2010*). Taken together with the broad range of specific phenotypes observed for human and murine mutations in the genes for individual COPII components, our findings suggest the potential for considerable heterogeneity in the COPII machinery, enabling the accommodation of diverse cargos.

Although complete deficiency for SEC24B or SEC24A in mice is compatible with embryonic development or survival to adulthood, respectively, no human patients have yet been identified with genetic

deficiencies in any of the four SEC24 paralogous genes. The SEC24A-deficient phenotype suggests a potential role for genetic variation at the *SEC24A* locus in the control of plasma cholesterol in humans, a key determinant of risk for myocardial infarction and stroke. Although genome-wide association studies for plasma lipid phenotypes have not identified a significant contribution from common genetic variants in the *SEC24A* gene (*Teslovich et al., 2010*), a role for rare alleles at this locus cannot be excluded. Complete deficiency of SEC24A, even in the presence of haplo-insufficiency of SEC24B, is compatible with survival and normal development in the mouse, suggesting that pharmacologic inhibition of hepatic SEC24A expression/function to achieve reduction in plasma cholesterol may be well tolerated as a potential approach to inhibit PCSK9 scretion.

## Materials and methods

### Mouse models and animal procedures

ES cell clone XE182 (129 genetic background) with a gene-trap insertion into intron 2 of the *Sec24a* gene, was obtained from the International Gene Trap Consortium (IGTC). The XE182 ES cell clone was expanded and then injected into C57BL/6J blastocysts at the University of Michigan Transgenic Mouse Core. Germ-line transmission was achieved by mating chimeric founders with 129/SvlmJ mice, and the resulting germ-line transmitted gene-trap allele was continuously backcrossed to C57BL/6J mice for at least seven generations prior to experimentation. Mice carrying a conditional *Sec24a* gene-trap allele in the C57BL6/J background were obtained from the Knockout Mouse Project (KOMP). *Sec24b* and *Sec24d* mutant mice have been described previously (*Baines et al., In press*; *Merte et al., 2010*). Genotyping was performed with mouse tail clip DNA using Go-Taq Green Master MIX (Promega, Madison, WI), and the resulting PCR products were resolved by 2% agarose gel electrophoresis. Primer sequences are listed under 'Primer sequences'. *Apoe* (stock no. 002052), *Apob* (stock no. 007682), *Ldlr* (stock no. 002207), and *Pcsk9* (stock no. 005993) mutant mice were obtained from The Jackson Laboratory. Transgenic mice (C57/BL6 background) carrying *Flpe* recombinase driven by an actin promoter (stock no. 005703) or *Cre* recombinase driven by an EIIA promoter (stock no. 003724) were obtained from the University of Michigan Transgenic Animal Core. Complete Blood Counts (CBC) were measured in an Advia120 whole blood analyzer (Bayer), according to the manufacturer's instructions. High fat diet (45% of Calories as fat) was purchased from Research Diets. Chow diet (TD 7001) supplemented with 60 mg/kg atorvastatin (Lipitor) was purchased from Harlan Teklad.

Animals were housed according to the guidelines of the University of Michigan Unit of Laboratory Animal Medicine (ULAM). Blood was collected using heparin-coated collection tubes (Fisher, Pittsburgh, PA) by retro-oribtal bleeding from mice anaesthetized with isoflurane. Fractionation of plasma samples and quantification of cholesterol were performed at the University of Cincinnati Mouse Metabolic Phenotyping Center. Adenovirus encoding *Cre* recombinase was purified and injected intravenously through mouse tail veins as previously described (*Li et al., 2008*). Plasma samples were then collected by centrifugation of heparinized blood samples at 3,000 g for 5 min at 4°C. Plasma cholesterol levels were measured with a colorimetric assay using the LiquiColor Cholesterol test kit (Stanbio, Boerne, TX). Plasma PCSK9 levels were determined using the mPCSK9 ELISA kit (Circulex, Woburn, MA), according to the manufacturer's instructions.

### DNA constructs

cDNAs of mouse *Sec24 a/b/c/d* were obtained from ATCC and sub-cloned into the pRFP-C1 vector (*Chen et al., 2007*) or the peGFP-C1 vector (Clontech, Mountain View, CA). Mouse *Sec23a* cDNA in the peGFP-C1 vector has been described previously (*Tao et al., 2012*). Mouse *Pcsk9* cDNA was purchased from the ATCC and subcloned into the pLenti vector (*Chen et al., 2007*), with or without a FLAG epitope fused at the C-terminus of PCSK9. A mammalian expression construct of Sar1 H79G was kindly provided by B. Ye (University of Michigan). Other constructs have been described previously (*Tao et al., 2012*). All constructs were confirmed by complete DNA sequencing at the University of Michigan DNA Sequencing Core.

### Non-denaturing SDS-PAGE and mass spectrometry

Ten µl plasma samples were first diluted with 90 µl Phosphate Buffer Saline (PBS; GIBCO, Grand Island, NY), and then mixed 1:1 with 2× SDS-PAGE sample buffer (Invitrogen, Grand Island, NY) without any reducing agent. 10 µl of each sample (equal to 0.5 µl plasma) was separated by 4–20% Tris-Glycine SDS-PAGE (Invitrogen) with the Seeblue Plus 2 protein size marker (Invitrogen). Gels

were first rinsed with de-ionized water for 10 min, and fixed with 20% methanol + 10% acetic acid for 30 min. Fixed gels were stained with 1% coomassie brilliant blue (Sigma-Aldrich, St Louis, MO) in 25% methanol for 60 min before de staining with 15% methanol for 15 min (3 times). HPLC-ESI-MS/MS was performed as described previously (*Yi et al., 2006*; *Chao et al., 2012*) with instrument specific modifications. Briefly, the gel portions containing APOA1 were excised, destained, dehydrated, and subjected to trypsin digestion overnight. The resulting peptides were desalted and analyzed by on-line HPLC on a Linear Trap Quadrupole-Orbitrap Elite (LTQ-Orbitrap Elite).

## Antibodies and recombinant proteins

An anti-SEC24A antibody was generated against a synthetic peptide (NTYDEIEGGGFLATPQL-C) in rabbits (Pacific Immunology, Ramona, CA). Purification of the rabbit serum was performed with an affinity column conjugated with the antigenic peptide as previously described (*Chen et al., 2011a*). Rabbit anti-FLAG antibody, mouse anti-actin antibody, and rabbit anti-SEC23 antibody were purchased from Sigma-Aldrich. Rabbit anti-PCSK9 (Ab-1) and rabbit anti-SREBP1 antibody were purchased from Santa Cruz Biotechnology. A second rabbit anti-PCSK9 antibody (Ab-2) was kindly provided by J Horton (UT Southwestern). Rabbit anti-APO-A1 antibody and rabbit anti-LMAN1 antibodies were purchased from Stressgen. Goat anti-albumin antibody, rabbit anti-SREBP2 antibody, rabbit anti-LDLR antibody, rabbit anti-Idol (Mylip) antibody, and HRP-conjugated anti-RFP antibody were purchased from Abcam (Eugene, OR). A second rabbit anti-Idol (Mylip) antibody was purchased from Protein Tech Group. Mouse anti-RALA antibody was purchased from BD Biosciences (Mountain View, CA). Preparation of anti-SAR1, anti-riboporin, anti-SEC24B, and anti-SEC22B antibodies, and purification of recombinant COPII proteins have been described previously (*Kim et al., 2007*).

## RNA extraction, high throughput mRNA-sequence (mRNA-Seq) analysis, and real-time PCR

Tissues were dissected from euthanized animals in cold PBS and immediately transferred to RNAlater (Ambion, Grand Island, NY) for storage. Total RNA was isolated using an RNeasy Kit (Qiagen). Whole transcriptome cDNA libraries were generated using TruSeq Stranded mRNA Sample Prep Kit (Illumina), sequenced on a Genome Analyzer II (Illumina, San Diego, CA), and sequencing data were mapped to the mouse reference genome mm9 using Bowtie (*Langmead et al., 2009*). Using ERANGE software (*Mortazavi et al., 2008*) (http://woldlab.caltech.edu/gitweb), the unique reads falling on the gene models and splice reads derived from splice junctions of genes were recorded. Following the author's recommendation, the unique reads were also re-evaluated by assessing a first-pass reads per KB per millions reads (RPKM), and the unique reads were subsequently re-calculated with weights computed during the first pass. All candidate regions that were within a 20 kb radius of a gene, a default parameter recommend by the script author, were reported. Final exonic read density (final RPKM) was determined according to the expanded exonic read density. Differential expression analysis was performed to compare wild type vs *Sec24a^{gt/gt}*. To obtain robust variance estimates, we empirically estimated the gene variance levels as a function of the average normalized read count per gene. Specifically, the gene-wise variance levels were estimated by calculating the differences in expression between samples for each gene, using local regression to predict the absolute value of the log-difference based on $\log_2$-read counts, and then estimating the variance using the standard sample variance formula and Fisher's Z distribution as in the methods of the eBayes function in the limma R package and in the IBMT method (*Smyth, 2004*; *Sartor et al., 2006*). This results in a moderated variance test that accounts for the dependence of variance on gene length and expression level. The variance estimates are expected to be slightly conservative, due to the assumption that relatively few genes are truly differentially expressed. Given the variance estimates, z-scores and p-values were calculated for each gene. Fold changes on the read counts (normalized per million bp) were calculated, and the p-values were adjusted for multiple testing using the Benjamini-Hochberg False Discovery Rate (FDR) approach (*Benjamini et al., 2001*). FDR < 0.05 and a fold change > 2 cut-off were used to select final up- and down-regulated gene lists. In addition to the gene level analysis, read counts and fold changes were also calculated using ERANGE for individual exons. Based on the counts of reads of individual exons and samples for each gene, Fisher's exact tests were performed to identify significant changes in expression ratios across exons, indicating differential splicing between the samples. Fisher's exact p-values were also adjusted for multiple testing using the Benjamini-Hochberg FDR approach. Reverse transcription was performed with the Superscript II cDNA Synthesis

Kit (Invitrogen). The iQ SYBR Green Supermix (ABI) was used for quantitative RT-PCR as previously described (*Li et al., 2008*). PCR primers are listed under 'Primer sequences.'

## Cell culture, transfection, and lentiviral gene transduction

293T and rat hepatoma McA-RH777 cells were grown in Dulbecco's Modified Eagle Medium (DMEM, GIBCO) containing 10% FBS (Sigma-Aldrich) and 1% Pen-Strip (GIBCO) at 37°C in the presence of 5% $CO_2$. 293T cells were transfected with Lipofectamine 2000 (Invitrogen) according to the manufacturer's instruction. McA-RH777 cells grown in six wells were first transfected with 100 nM siRNA pools (equal mixture of four different siRNAs; sequences are listed in 'Primer Sequences') at ~50% confluence. Cells were split 48 hr later to 60 mm dishes to ~50% confluency for another round of siRNA treatment. Cytosol was prepared 48 hr after the third round of siRNA treatment. Production of lentivirus was carried out in 293T cells essentially as described (*Chen et al., 2011a*). Infection was performed by spinning viral particles onto cells at 1000 g for 45 min at room temperature in the presence of 8 µg/ml polybrene.

For experiments involving conditioned medium collection, cells were washed once with serum free DMEM medium and then maintained in serum free DMEM for 4 hr before harvest. Conditioned medium was then centrifuged at 3,000 g for 5 min at 4°C to sediment floating cells or debris. Brefeldin A was purchased from Cell Signaling and dissolved in ethanol and used at a final concentration of 1 µg/ml.

## Generation of permeabilized cells and in vitro budding and biochemical procedures

In vitro budding experiments using permeabilized cells were performed as previously described (*Kim et al., 2007*) with slight modification. Three plates of 293T cells stably expressing PCSK9-FLAG grown to ~80–90% confluency were collected by trypsinization and sedimented at 750 g for 5 min at 4°C. Cells were resuspended in 6 ml of B88 buffer (20 mM HEPES pH 7.2, 250 mM sorbitol, and 150 mM KOAc) and permeabilized with 40 µg/ml digitonin on ice for 5 min. Permeabilization was stopped by adding ice-cold 8 ml of B88 buffer and permeabilized cells were sedimented at 750 g for 5 min at 4°C. After washing twice with 6 ml B88 buffer at 4°C, permeabilized cells were resuspended in 1 ml B88 buffer and centrifuged at 10,000 g for 1 min at 4°C. Permeabilized cells were then resuspended in 1 ml B88 buffer supplemented with 1 M $MgCl_2$ for 10 min on ice to wash off cytosolic protein associated with cellular membranes. Permeabilized cells were then washed three times with 1 ml B88 buffer after being centrifuged at 10,000 g for 20 s, and finally resuspended in 0.2–0.3 ml B88 buffer. Budding reactions (100 µl) were assembled in non-stick eppendorf tubes on ice with 20 µl (OD600 = 0.1–0.2) semi-inact cells as donor membranes, 10 µl of 10× ATP regeneration system (10 mM ATP, 400 mM creatine phosphate, 2 mg/ml creatine phosphokinase, and 5 mM MgOAc in B88 buffer), 1.5 µl of 10 mM GTP, and rat liver cytosol at 4 mg/ml final concentration. For budding experiments with cytosols from McA-RH777 cells, the final concentration of cytosols was 2 mg/ml. Reactions were performed at 30°C for 60 min and stopped by centrifuging at 10,000 g for 10 min at 4°C. 75 µl supernatant was centrifuged in a TLA100 rotor at 100,000 g for 10 min at 4°C. The supernatants were discarded by pipetting with gel loading tips, and the high speed pellet fractions (include COPII vesicles) were thoroughly resuspended in 20 µl of 1× SDS sample buffer (Invitrogen) supplemented with 5% β-mercaptoethanol (β-ME) and heated at 55°C for 20 min before SDS-PAGE.

Proteins were extracted from tissues or cells at 4°C for 60 min with buffer A (100 mM Tris pH 7.5, 1% NP-40, 10% glycerol, 130 mM sodium chloride, 5 mM magnesium chroride, 1 mM sodium vanadate, 1 mM sodium fluoride, and 1 mM EDTA) supplemented with protease inhibitor tablets (Roche), according to the manufacturer's instructions. Supernatants were collected after spinning at 13,000 g at 4°C for 10 min, and mixed 1:1 with 2× SDS sample buffer (Invitrogen) with 10% β-ME (Sigma-Aldrich). For endoglycosidase H (Endo H) treatment, 20 µg of protein in 1× SDS sample buffer was incubated with 10U of Endo H (NEB) at 37°C for 2 hr. The enzyme was inactivated by heating at 70°C for 20 min. 10–20 µg of protein was loaded on 4–12% Tris-Glycine SDS-PAGE for separation. Protein extracts were incubated with 20 µl wheat germ agglutinin (WGA) beads (EY Laboratories, San Mateo, CA) for 4 hr at 4°C before washing four times with buffer A. Protein bound to WGA beads were solubilized with 1× SDS-PAGE sample buffer (Invitrogen) with 5% β-ME, and separated with 4–12% Tris-glycine SDS-PAGE (Invitrogen). Immunoprecipitation was carried out with cellular proteins extracted at 4°C for 30 min with CHAPS based buffer B (*Kim et al., 2002*) (100 mM Tris pH 7.5, 0.5% CHAPS, 10% glycerol, 130 mM sodium chloride, 10 mM sodium pyrophosphate, 5 mM magnesium chloride, 1 mM sodium fluoride, and 1 mM EDTA) supplemented with protease inhibitor tablets (Roche). Cell lysates were then incubated with 10 µl M2 agarose (Sigma-Aldrich) for 4 hr at 4°C before washing four times with buffer B.

Immune-complexes were then solubilized with 1X SDS-PAGE sample buffer (Invitrogen) with 5% β-ME, and separated with 4–12% Tris-glycine SDS-PAGE (Invitrogen). Following SDS-PAGE, proteins were transferred onto nitrocellulose membranes (Bio-rad) and immunoblotting was performed as previously described (*Chen et al., 2011b*).

## Primer sequences

*Sec24a*-F (5′-GGGTAAGAGCAGCACCCGACTG)
 *Sec24a*-R (5′-ATGTGCCCTAGGCATGAAAC)
 *Sec24a*-V (5′-GGGTCTCAAAGTCAGGGTCA)
 *Sec24a*-A (5′-CTGTCTTACAGGTTGTTCCGATGCACGCTG)
 *Sec24a*-B (5′-CACAGCCAGCCCAGTGGTAT)
 *Sec24a*-C (5′-AGGAAAAGAACCCTGTCATA)
 *Sec24a*-C (5′-CACACCTCCCCCTGAACCTGAAAC)
 *Sec24a*-Exon2 (5′-CAATATGTTTCTTCTGGAGACCC)
 *Sec24a*-Exon3 (5′-AGCTGAGTTAAATGAAGGTGGGG) *Hmgcr* (5′-ATGTTCACCGGCAACAACAA, 5′-GCGATGCACCGCGTTATC) *Ldlr* (5′-CATCCGAGCCATTTTCACAGTC, 5′-CTGACTTGTCCTTGCA-GTCTGC) *Scd1* (5′-GCTGGAGTACGTCTGGAGGAA, 5′-TCCCGAAGAGGCAGGTGTAG) *Mttp* (5′-AGGCCGTCCAGAGCTTCCTG, 5′-GAGTCTGAGCAGAGGTGACG) *Apob* (5′-GAGTTCCAGATG-GTGTCTCCAAG, 5′-CTTGGAGTCTGACAAAGCTTAGC) *Apoe* (5′-CAGACGCTGTCTGACCAGGTC, 5′-GTGTCTCCTCCGCCACTGGAC) *Pcsk9* (5′-AGTTGCCCCATGTGGAGTACA, 5′-TCTGGGCGAAGACA AAGGAGT)

Stealth RNAi against rat *Sec24a/b* sense sequences:
 *rSec24a*-1: UCGUUUCAGGUAAUCCUCAAAGAUU
 *rSec24a*-2: CCUAUCCCACCCGAUCGACUCUAAA
 *rSec24a*-3: CGGUCUGUCAAGAAGGUGACGUUCU
 *rSec24a*-4: GCUUUCCUGUUGGAGCUCUUAGGAU
 *rSec24b*-1: CAUGGAAUGACAUGUCACAAUCUAA
 *rSec24b*-2: CCAGAUAAAGCCAUCGCACAGUUAA
 *rSec24b*-3: CCACAGUACUCGUCUGUAUGAUUUA
 *rSec24b*-4: CAGUUGGUUUGGUGGUUCGUUUGUU

## Acknowledgements

The authors would like to thank Dr Zhouji Chen for insightful comments and suggestions and Dr Jay Horton for sharing the anti-PCSK9 antibody.

## Additional information

### Competing interests

SGY, DG: Reviewing editor, *eLife*. RS: Editor-in-Chief, *eLife*. The other authors declare that no competing interests exist.

### Funding

| Funder | Grant reference number | Author |
| --- | --- | --- |
| Howard Hughes Medical Institute | | Kanika Bajaj, Randy Schekman, David Ginsburg |
| National Institutes of Health | P01HL057346, R01HL039693 | Xiao-Wei Chen, He Wang, Zhuo-Xian Meng, Elizabeth Adams, Andrea Baines, Bin Zhang, Jiandie Lin, Stephen G Young, David Ginsburg |
| American Heart Association | 10POST4150049 | Xiao-Wei Chen |

The funders had no role in study design, data collection and interpretation, or the decision to submit the work for publication.

## Author contributions

X-WC, Conception and design, Acquisition of data, Analysis and interpretation of data, Drafting or revising the article; HW, BZ, Conception and design, Acquisition of data, Analysis and interpretation of data; KB, Conception and design, Analysis and interpretation of data; PZ, Analysis and interpretation of data, Drafting or revising the article; Z-XM, Conception and design, Acquisition of data; DM, YB, ZY, Acquisition of data, Analysis and interpretation of data; H-HL, GY, Acquisition of data, Drafting or revising the article; EA, AB, Acquisition of data, Contributed unpublished essential data or reagents; MAS, SGY, RS, DG, Conception and design, Analysis and interpretation of data, Drafting or revising the article; JL, Acquisition of data, Analysis and interpretation of data, Drafting or revising the article

## Ethics

Animal experimentation: The University Committee of Use and Care of Animal (UCUCA) of the University of Michigan approved animal protocol number is 08571. The institutional guidelines for the care and use of laboratory animals were followed in all experiments involving animals.

# Additional files

## Supplementary files

• Supplementary file 1. RNA-Seq ENRAGE results. (**A**) Genes. (**B**) Exons.

• Supplementary file 2. Gene-ontology analysis of potentially biological pathways in *Sec24a*[gt/gt] mice. (**A**) Down-regulated. (**B**) Up-regulated.

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
