## [Decision Letter]

Thank you for choosing to send your work entitled “SEC24A Deficiency Lowers Plasma Cholesterol through Reduced PCSK9 Secretion” for consideration at *eLife*. Your article has been favorably evaluated by a Senior editor and 3 reviewers, one of whom, Helen Hobbs, is a member of our Board of Reviewing Editors.

The Reviewing editor and the other two reviewers discussed their comments before we reached this decision, and the Reviewing editor has assembled the following comments to help you prepare a revised submission:

All three reviewers consider your paper suitable for publication in *eLife* after appropriate revisions have been made. The paper is of general interest to those in the secretory and lipoprotein metabolism fields. A summary of the reviewers' comments follows, which includes the experiments that need to be done for the paper to be accepted. Other comments and suggestions are provided that you may want to address, but these are not required for acceptance. We do not anticipate that the paper will be sent out for re-review.

**General comments**

The authors show that inactivation of *Sec24a* in mice using two different targeting constructs has no apparent effect on viability or development (up to 12 months), but results in a strikingly restricted phenotype, hypocholesterolemia. The authors provide evidence that this surprising result is caused by a reduction in secretion of PCSK9, a protein that binds the extracellular domain of the LDL receptor and promotes receptor degradation. The authors demonstrate convincingly that the knockout mice have reduced levels of circulating PCSK9 coupled with increased levels of both the ER-form of PCSK9 and of the LDLR in the liver. This is all consistent with the hypocholesterolemia in the *Sec24a*^*-/-*^ mice being due to a deficiency in PCSK9 secretion, as hypothesized by the authors.

*In vitro* biochemical analyses reveal that SEC24A, but not other SEC24 isoforms, co-precipitate PCSK9 when the proteins are overexpressed in 293 cells and that overexpression of SEC24A enhances secretion of overexpressed PCSK9. These studies, combined with studies of other SEC24 isoform knockout mice, demonstrate differential requirements for COPII subunits in cargo selection at the ER. The relative selectivity of SEC24A for protein secretion is remarkable and of general interest. Since PCSK9 is a soluble cargo molecule, these data suggest that a PCSK9 cargo receptor may exist.

**Substantive concerns to be addressed during revision**

1) Although evidence is provided that PCSK9 is packaged in COPII vesicles using an *in vitro* COPII vesicle budding assay, the authors need to show a direct mechanistic link between SEC24A deficiency and PCSK9 secretion. This could be shown by knocking down SEC24A in the cells used in the COPII vesicle budding assay to determine if there is a selective reduction in PCSK9 trafficking. A more direct test would be to perform budding assay using mouse liver cytosol from WT versus *Sec24a*^*-/-*^ mice.

Other mechanisms affecting LDLR levels should be ruled out. For example, are levels of the LDLR E3 ligase Idol altered in *Sec24A* knockout mice?

2) The quality of the PCSK9 blots on liver tissue do not permit assessment of the relative amounts of the full length versus cleaved PCSK9 (for example, see S Rashid et al. PNAS 2005). Even in the 293 cells stably expressing PCSK9, the mature form of the protein is not visible on immunoblotting due to the presence of a cross-reactive band. This problem could be avoided by using another anti-PCSK9 Ab.

3) The therapeutic potential of inactivating SEC24A should be de-emphasized. The suggestion that SEC24A may be a target for LDL lowering is premature, especially given the fact that there are phenotypic effects of inactivating *Sec24A* that are not due to a deficiency of PCSK9, including mild increases in RBC MCV and MCHC, and a >50% reduction in SCD1 mRNA. There are likely to be others.

---

## [Author Response]

*1) Although evidence is provided that PCSK9 is packaged in COPII vesicles using an* in vitro *COPII vesicle budding assay, the authors need to show a direct mechanistic link between SEC24A deficiency and PCSK9 secretion. This could be shown by knocking down SEC24A in the cells used in the COPII vesicle budding assay to determine if there is a selective reduction in PCSK9 trafficking. A more direct test would be to perform budding assay using mouse liver cytosol from WT versus* Sec24a^-/-^
*mice*.

As suggested by the reviewers, we performed a new series of *in vitro* COPII reconstitution assays using cytosol prepared from rat hepatoma McA-RH777 cells in which SEC24A, SEC24B, or both were depleted by siRNA treatment. The deficient cytosols exhibited decreased activity in the generation of PCSK9-containing COPII vesicles, most marked in the SEC24A/B double knockdown experiments. These data have now been added as Figure 8d–f in the revised manuscript. A similar reduction in the efficiency of packaging of SEC22B (a SNARE protein) was observed (Figure 8e), however, LMAN1 packaging was not affected. PCSK9 and SEC22B may share a requirement for SEC24A/B whereas LMAN1 was shown as a non-selective COPII cargo. The LMAN1 result also demonstrates that the other COPII proteins were not depleted from the SEC24A/ B knockdown cytosol.

As suggested by the reviewers, we also attempted similar *in vitro* budding assays using mouse liver cytosol prepared from control and knockout mice. Unfortunately, both mouse liver cytosols failed to exhibit any activity in the COPII vesicle budding assay, even for the classic COPII cargos LMAN1 and SEC22. A similar lack-of-activity was observed using cytosols prepared from mouse brain. We hypothesize that these results reflect the loss of one or more key components during cytosol preparation from mouse tissues, or the presence in these preparations of an inhibitor to the budding reaction. Our anti-SEC24A antibody also failed to exhibit blocking activity in the budding assay. These data are included below:

*Other mechanisms affecting LDLR levels should be ruled out. For example, are levels of the LDLR E3 ligase Idol altered in Sec24A knockout mice*?

Based on the reviewers' suggestion, we examined IDOL protein levels by immunoblotting using 2 different anti-IDOL antibodies. No differences in IDOL were observed between liver samples prepared from SEC24A-deficient mouse or their wild type littermates. These data have been added to Figure 6c in the revised manuscript.

*2) The quality of the PCSK9 blots on liver tissue do not permit assessment of the relative amounts of the full length versus cleaved PCSK9 (for example, see*
*S Rashid et al. PNAS 2005**). Even in the 293 cells stably expressing PCSK9, the mature form of the protein is not visible on immunoblotting due to the presence of a cross-reactive band. This problem could be avoided by using another anti-PCSK9 Ab*.

As suggested by the reviewers, we obtained the PCSK9 antibody referenced above (courtesy of JD Horton, UT Southwestern Medical Center). Immunoblotting using this reagent demonstrated that both mature PCSK9 and pro-PCSK9 are increased in liver lysates from SEC24A-deficient mice compared to their wild-type littermate controls. These data have been added to Figure 7a in the revised manuscript.

*3) The therapeutic potential of inactivating SEC24A should be de-emphasized. The suggestion that SEC24A may be a target for LDL lowering is premature, especially given the fact that there are phenotypic effects of inactivating Sec24A that are not due to a deficiency of PCSK9, including mild increases in RBC MCV and MCHC, and a >50% reduction in SCD1 mRNA. There are likely to be others*.

This point is well taken. In the revised manuscript, we have substantially de-emphasized the potential of SEC24A as a therapeutic target. However, it is interesting to note that in the original report of the PCSK9 knockout mouse (S Rashid et al PNAS 2005), a >50% reduction in SCD1 mRNA was also observed, similar to what we observe in SEC24A-deficient mice.